

# Influence of ambient NO and NO₂ on the quantification of total peroxy nitrates (∑PNs) and total alkyl nitrates (∑ANs) by thermal dissociation cavity ring-down spectroscopy (TD-CRDS)

Laura Wüst[1], Patrick Dewald[1], Gunther N. T. E. Türk[1], Jos Lelieveld[1] and John N. Crowley[1]

[1]Atmospheric Chemistry Department, Max Planck Institute for Chemistry, Mainz, 55128, Germany

*Correspondence to*: John N. Crowley (john.crowley@mpic.de)

**Abstract**

Measurement of total peroxy nitrates (∑PNs) and alkyl nitrates (∑ANs) by instruments that use thermal dissociation (TD) inlets

to convert the organic nitrate to detectable $NO_2$ may suffer from systematic bias (both positive and negative) resulting from unwanted secondary chemistry in the heated inlets. Here we review the sources of the bias and the methods used to reduce it and/or correct for it and report new experiments using (for the first time) atmospherically relevant, unsaturated, biogenic alkyl nitrates as well as two different peroxyacetyl nitrate (PAN) sources. We show that the commonly used commercial C3-alkyl-nitrate (isopropyl nitrate, IPN) inlet for characterising the chemistry of ANs is not appropriate for real-air samples that contain

longer chain nitrates. ANs generated in the $NO_3$-induced oxidation of limonene are strongly positively biased in the presence of NO. By detecting $NO_X$ rather than $NO_2$, we provide a simple solution to avoid the bias caused by the conversion of NO to $NO_2$ by primary and secondary peroxy radicals resulting from the complex chemistry in the thermal degradation of long-chain, alkyl nitrates in air at TD-temperatures. We also show that using a photochemical source of PAN to characterise the TD-inlets can result in a much stronger apparent bias from NO to $NO_2$ conversion than for a diffusion source of synthesised ("pure")

PAN at similar mixing ratios. This is explained by the presence of thermally labile trace gases such as peracetic acid ($CH_3C(O)OOH$) and hydrogen peroxide ($H_2O_2$).

## 1 Introduction

Nitrogen monoxide (NO) and nitrogen dioxide ($NO_2$), together known as $NO_X$, have a significant impact on air quality and human health (Crutzen, 1979; Lelieveld et al., 2015). Measurements of trace gases that act as $NO_x$ reservoirs or sinks are

essential to gain a deeper insight into the degradation and transport mechanisms of $NO_X$. Organic compounds with nitrate functionality, such as alkyl (aliphatic) nitrates (ANs, $RONO_2$) and peroxy nitrates (PNs, $RO_2NO_2$), influence the lifetime and transport of $NO_X$ in the atmosphere (Horowitz et al., 2007; Perring et al., 2013) and represent a large fraction of the tropospheric nitrogen reservoir (Roberts, 1990). PNs are formed by the reaction between organic peroxy radicals ($RO_2$) and $NO_2$ (R1), while ANs are formed by the reaction between $RO_2$ and NO (R2) in competition with the formation of alkoxy radicals (R3), which

is usually the dominant reaction pathway (Orlando and Tyndall, 2012). PNs with a carbonyl group on the α-carbon, such as



PAN, have lifetimes with respect to thermal dissociation to $NO_2$ of minutes to hours in the boundary layer, extending to years in the cold upper troposphere (Atkinson et al., 2006). In contrast, $RO_2NO_2$, which do not possess a α-carbonyl group decompose within seconds in the temperate boundary layer and these PNs have been detected only at ground level in polar regions (Slusher et al., 2002) or at high altitudes (Murphy et al., 2004).

$$RO_2 + NO_2 + M \rightarrow RO_2NO_2 + M \quad\quad\quad (R1)$$
$$RO_2 + NO + M \rightarrow RONO_2 + M \quad\quad\quad (R2)$$
$$RO_2 + NO \rightarrow RO + NO_2 \quad\quad\quad (R3)$$

Organic peroxy radicals are formed during the day by the OH-initiated oxidation of volatile organic compounds (VOCs) (R4) (Lightfoot et al., 1992). During night, OH levels are much lower and the $O_3$-initiated oxidation of VOCs (R5) gains importance
as a source of $RO_2$ (Ng et al., 2017).

$$VOC + OH + (O_2) \rightarrow RO_2 + products \quad\quad\quad (R4)$$
$$VOC + O_3 + (O_2) \rightarrow RO_2 + products \quad\quad\quad (R5)$$

Nitrate radicals ($NO_3$) and dinitrogen pentoxide ($N_2O_5$) play important roles in atmospheric nitrogen chemistry, especially at nighttime (Brown and Stutz, 2012). The reaction of $NO_2$ with $O_3$ produces the $NO_3$ radical (R6), which is in thermal
equilibrium with $NO_2$ and $N_2O_5$ (R7) (Wayne et al., 1991).

$$NO_2 + O_3 \rightarrow NO_3 + O_2 \quad\quad\quad (R6)$$
$$NO_2 + NO_3 + M \rightleftharpoons N_2O_5 + M \quad\quad\quad (R7)$$

During the day, $NO_3$ is rapidly photolysed (Liebmann et al., 2018) or reacts with NO to form $NO_2$ (Wayne et al., 1991). At night, however, $NO_3$ can reach concentrations of up to several tens of pptv (Heintz et al., 1996; Geyer et al., 2001; Brown et
al., 2003; Dewald et al., 2022) and initiates the oxidation of many unsaturated VOCs (R8), especially terpenoids (Ng et al., 2017). In environments with high levels of biogenic VOCs, $NO_3$ also contributes to their oxidation during the day (Liebmann et al., 2019; Dewald et al., 2024).

$$NO_3 + VOC + (O_2) \rightarrow RONO_2 + products \quad\quad\quad (R8)$$

PNs and ANs have often been measured using chromatographic (Flocke et al., 2005) and mass spectrometric methods
(Slusher et al., 2004; Zheng et al., 2011; Wu et al., 2021) that allow identification and quantification of the individual components. An alternative method for measuring total, non-speciated PNs and ANs (i.e. $\sum$PNs and $\sum$ANs) is thermal dissociation to $NO_2$ (R9-R10) and subsequent detection by cavity ring-down spectroscopy (TD-CRDS) or laser induced fluorescence (TD-LIF) (Day et al., 2002; Paul et al., 2009; Paul and Osthoff, 2010; Sobanski et al., 2016; Thieser et al., 2016).

$$RO_2NO_2 + M \rightarrow RO_2 + NO_2 + M \quad\quad\quad (R9)$$
$$RONO_2 + M \rightarrow RO + NO_2 + M \quad\quad\quad (R10)$$

This method makes use of the very different bond-strengths of $ROO-NO_2$ and $RO-NO_2$ to separate their detection in inlets at appropriate temperatures (Day et al., 2002), typically around 473 K for PNs and 673 K for ANs for instruments developed in





this group for example (Sobanski et al., 2016; Thieser et al., 2016). As discussed below, the thermal dissociation of PNs and

ANs to NO$_2$ can however result in deviation from ideal (1:1) conversion as a result of unwanted, secondary chemistry.

## 2 Sources of potential bias in TD-CRDS measurements of PNs and ANs

The quantitative measurement of ∑ANs and ∑PNs as NO$_2$ following thermal dissociation can be influenced by secondary

reactions that either generate or consume the NO$_2$ product. The origin of the bias and the various methods that have been used

to reduce it or correct for it are summarised below.

The three types of reactions that can modify the concentration of NO$_2$ formed in thermal dissociation of ∑ANs or ∑PNs are

its reduction to NO (R11-R12),  the oxidation of ambient NO to NO$_2$ via reaction with O$_3$ or with RO$_2$ formed in the TD-inlet

(R3, R13) and the conversion of NO$_2$ back to an organic/inorganic nitrate (R14, R15) (Day et al., 2002; Paul et al., 2009; Paul

and Osthoff, 2010; Di Carlo et al., 2013; Sobanski et al., 2016; Thieser et al., 2016).

$$O_3 + M \quad\quad \rightarrow \quad\quad O(^3P) + O_2 + M \quad\quad\quad\quad (R11)$$

$$O(^3P) + NO_2 \quad\quad \rightarrow \quad\quad NO + O_2 \quad\quad\quad\quad (R12)$$

$$O_3 + NO \quad\quad \rightarrow \quad\quad O_2 + NO_2 \quad\quad\quad\quad (R13)$$

$$RO_2 + NO \quad\quad \rightarrow \quad\quad RO + NO_2 \quad\quad\quad\quad (R3)$$

$$OH + NO_2 + M \quad\quad \rightarrow \quad\quad HNO_3 + M \quad\quad\quad\quad (R14)$$

$$RO_2 + NO_2 + M \quad\quad \rightarrow \quad\quad RO_2NO_2 + M \quad\quad\quad\quad (R15)$$

The size of the bias caused by these reactions depends sensitively on characteristics of the heated inlet, in particular the pressure

at which it is operated. Instruments that use CRDS setups to detect NO$_2$ are usually operated at cavity pressures (and thus inlet

pressures) above 500 hPa, whereas the TD-inlets of instruments using LIF to detect NO$_2$ can be operated at any pressure above

that of the LIF-module, which is typically at 3-5 hPa (Day et al., 2002). Operation at low pressures is generally preferred as it

enhances the rate of diffusive loss of radicals to the inlet walls, minimising the bias. A potential disadvantage of operation at

low temperatures is the enhancement of the O($^3$P) concentration as its (termolecular) reaction with O$_2$ (the reverse of R11) to

reform O$_3$ is greatly reduced (Lee et al., 2014). Recognising that many of the secondary processes involve radicals and atoms

formed in the heated inlets, Sobanski et al. (2016) added glass beads in the TD-section of the inlet of their CRD-based device

to scavenge peroxy and OH radicals. While this successfully reduced the impact of O($^3$P), OH and RO$_2$ radical reactions

(reducing the observed bias), the use of glass beads with a large surface area was later shown to reduce the temperature at

which unsaturated (e.g. biogenic) ANs are converted to NO$_2$ in the presence of O$_3$, resulting in the partial detection of ANs in

the PNs channel with an inlet at 473 K (Dewald et al., 2021). This effect was weaker, but still apparent when using a quartz

inlet without glass beads. This observation led Dewald et al. (2021) to switch to a Teflon (PFA) inlet which does not support

the formation of reactive surface sites (S-O) that calalyse the multistep conversion of ANs to NO$_2$. Note that such effects are

not observed for saturated ANs (e.g. isopropyl nitrate, which has often been used as easily available "standard" to test inlet

effects), which do not react with O$_3$.



An alternative approach to the physical reduction of bias (e.g. by removing radicals in the TD-inlet) is a correction based on laboratory characterisation of the effects of adding $O_3$, NO and $NO_2$ to the inlets in the presence of PNs and ANs followed by numerical simulation of the results using reaction schemes that describe the gas-phase and heterogeneous reactions taking place (Sobanski et al., 2016; Thieser et al., 2016).

Table 1 presents a summary of the bias measured by various groups using TD-inlets coupled to various $NO_2$ detection schemes and the approaches used to minimise and /or correct them.

**Table 1: Overview of the bias from secondary chemistry occuring when using TD-inlets and various detection methods for NO₂.**

| Process | TD-Inlet | Method | Organic Nitrate | Bias | Size of bias | Corrective approach | Reference |
|---|---|---|---|---|---|---|---|
| $O(^3P) + NO_2$ | >603 K | LIF | -- | -ve | < 3.5 % loss of $NO_2$ (200 – 240 ppbv $O_3$) | -- | Day et al. (2002) |
| $O(^3P) + NO_2$ | 653 K | LIF | -- | -ve | 6 % loss of $NO_2$ (30 ppbv $O_3$) | a) Expression based on laboratory data | Lee et al. (2014) |
| $O(^3P) + NO_2$ | 723 K | CRDS | -- | -ve | < 4 % loss of $NO_2$ (50 ppbv $O_3$) | b) Expression based on laboratory data | Thieser et al. (2016) |
| $O(^3P) + NO_2$ | 648 K | CRDS | -- | -ve | 1.5 % loss of $NO_2$ (50 ppbv $O_3$) | Glass beads, numerical simulation | Sobanski et al. (2016) |
| $O(^3P) + NO_2$ | 653 K | CEAS | -- | -ve | Not significant | -- | Li et al. (2021) |
| | | | | | | | |
| $O_3 + NO$ | 873 K | LIF | -- | +ve | 1.6 % of the NO converted to $NO_2$ (100 ppbv $O_3$) | Bias quantified in laboratory data | Day et al. (2002) |
| $O_3 + NO$ | 673 K | LIF | -- | +ve | < 1 % of the NO | Reduced pressure (< 5 HPa), numerical simulation. | Wooldridge et al. (2010) |
| $O_3 + NO$ | 473 K 723 K | CRDS | -- | +ve | 0.13 ppbv (1 ppbv NO and 50 ppbv $O_3$) 0.33 ppbv (1 ppbv NO and 50 ppbv $O_3$) | c) Expression based on laboratory data | Thieser et al. (2016) |
| $O_3 + NO$ | 448 K 648 K | CRDS | -- | +ve | 1.0 ppbv (1 ppbv NO and 50 ppbv $O_3$) 1.7 ppbv (1 ppbv NO and 50 ppbv $O_3$) | Glass beads, numerical simulation | Sobanski et al. (2016) |
| $O_3 + NO$ | 453 K 653 K | CEAS | -- | +ve | Not significant (within the measurement uncertainty) | -- | Li et al. (2021) |
| | | | | | | | |
| $RO_2 + NO$ | 523 K | CRDS | PAN | +ve | Factor 1.20 (1 ppbv PAN and 4 ppbv NO) | Second order polynomial expression based on laboratory data | Paul and Osthoff (2010) |
| $RO_2 + NO$ | 473 K | LIF | PAN | +ve | < Factor 1.05 (NO < 3 ppbv) | Reduced pressure (< 5 HPa), numerical simulation. | Wooldridge et al. (2010) |
| $RO_2 + NO$ | 463 K | CIMS | PPN | -ve | Factor 2 (2.2 ppbv PPN and 5 ppbv NO) | Laboratory data. Suppression of ion signal from matrix effects. | Mielke and Osthoff (2012) |
| $RO_2 + NO$ | 473 K | CRDS | PAN | +ve | Factor 1.80 (1 ppbv PAN and 4 ppbv NO) | Numerical simulation | Thieser et al. (2016) |
| $RO_2 + NO$ | 723 K | CRDS | PAN | +ve | Factor 1.25 (1 ppbv PAN and 4 ppbv NO) | Numerical simulation | Thieser et al. (2016) |
| $RO_2 + NO$ | 448 K | CRDS | PAN | +ve | Factor 1.40 (3 ppbv PAN and 8 ppbv NO) | Glass beads, numerical simulation | Sobanski et al. (2016) |
| $RO_2 + NO$ | 648 K | CRDS | PAN | +ve | Factor 1.04 (3 ppbv PAN and 8 ppbv NO) | Glass beads, numerical simulation | Sobanski et al. (2016) |
| $RO_2 + NO$ | 433 K 633 K | CAPS | -- | +ve | Not mentioned | -- | Sadanaga et al. (2016) |
| $RO_2 + NO$ | 523 K | PERCA-CRDS | PAN | +ve | Factor 4.0 (2 ppbv PAN and 0.75 ppmv NO) | Numerical simulation | Taha et al. (2018) |
| $RO_2 + NO$ | 403 K | CAPS | -- | +ve | Not determined | Numerical simulation | Keehan et al. (2020) |
| $RO_2 + NO$ | 453 K | CEAS | PAN | +ve | Factor 1.56 (2.8 ppbv PAN and 5 ppbv NO) | Numerical simulation | Li et al. (2021) |
| $RO_2 + NO$ | 653 K | CEAS | PAN | +ve | Factor 1.40 (2.8 ppbv PAN and 5 ppbv NO) | Numerical simulation | Li et al. (2021) |
| $RO_2 + NO$ | 453 K | CRDS | PAN | +ve | Factor 1.12 (2.11 ppbv PAN and 5 ppbv NO) | Glass beads | Lin et al. (2024) |
| $RO_2 + NO$ | 633 K | CRDS | PAN | +ve | Not significant | Glass beads, laboratory data | Lin et al. (2024) |
| $RO_2 + NO$ | 433 K | CAPS | PAN | +ve | Factor 4.10 (0.8 ppbv PAN and 8.5 ppbv NO) | Detection of $NO_X$ | Ohara et al. (2024) |
| $RO_2 + NO$ | 448 K | CRDS | PAN | +ve | Factor 1.55 (2 ppbv PAN and 4 ppbv NO). | Detection of $NO_X$ | This work |
| $RO_2 + NO$ | 648 K | CRDS | PAN | +ve | Factor 1.70 (2 ppbv PAN and 4 ppbv NO). | Detection of $NO_X$ | This work |
| $RO_2 + NO$ | 673 K | LIF | NPN | +ve | 0.7 % conversion of NO to $NO_2$ per ppbv of AN | Reduced pressure (2.7-5.3 HPa) | Day et al. (2002) |
| $RO_2 + NO$ | 723 K | CRDS | IPN | +ve | 1.6 % conversion of NO to $NO_2$ per ppbv of AN | Numerical simulation | Thieser et al. (2016) |
| $RO_2 + NO$ | 648 K | CRDS | IPN | +ve | Factor 1.2 (1.4 ppbv IPN and 6 ppbv NO) | Glass beads, numerical simulation | Sobanski et al. (2016) |
| $RO_2 + NO$ | 653 K | CEAS | MN | +ve | Not mentioned | Numerical simulation | Li et al. (2021) |
| $RO_2 + NO$ | 633 K | CRDS | EHN | +ve | Factor 1.12 (3.36 ppbv EHN and 6 ppbv NO) | Glass beads, standard addition experiments | Lin et al. (2024) |
| $RO_2 + NO$ | 633 K | CAPS | IPN | +ve | Factor 2.8 (0.97 ppbv IPN and 6 ppbv NO) | Detection of $NO_X$ | Ohara et al. (2024) |



| RO₂ + NO | 648 K | CRDS | Limonene-nitrate | +ve | 25 % conversion of NO to NO₂ per ppbv of AN | Detection of NOₓ | This work |
|---|---|---|---|---|---|---|---|
| RO₂ + NO | 648 K | CRDS | Isoprene-nitrate | +ve | 5.1 % conversion of NO to NO₂ per ppbv of AN | Detection of NOₓ | This work |
| | | | | | | | |
| RO₂ + NO₂ | 523 K | CRDS | PAN | -ve | Not significant | -- | Paul et al. (2009) |
| RO₂ + NO₂ | 473 K | LIF | PAN | -ve | > Factor 1.05 (NOₓ < 3 ppbv) | Simulations and operate at a pressure of < 5 Hpa | Wooldridge et al. (2010) |
| RO₂ + NO₂ | 463 K | CIMS | PPN | -ve | Factor 1.25 (2.2 ppbv PPN and 5 ppbv NO) | Laboratory data. Suppression of ion signal from matrix effects | Mielke and Osthoff (2012) |
| RO₂ + NO₂ | 473 K | LIF | PAN | -ve | 3 pptv per ppbv NO₂ | | Di Carlo et al. (2013) |
| RO₂ + NO₂ | 473 K | CRDS | PAN | -ve | Factor 1.52 (0.7 ppbv PAN and 8 ppbv NO₂) | Numerical simulation | Thieser et al. (2016) |
| RO₂ + NO₂ | 723 K | CRDS | PAN | -ve | Not significant | -- | Thieser et al. (2016) |
| RO₂ + NO₂ | 448 K | CRDS | PAN | -ve | Factor 1.06 (9 ppbv PAN and 8 ppbv NO₂) | Numerical simulation | Sobanski et al. (2016) |
| RO₂ + NO₂ | 648 K | CRDS | PAN | -ve | Not significant | -- | Sobanski et al. (2016) |
| RO₂ + NO₂ | 403 K | CRDS | Carene-nitrate | -ve | Factor 2.33 (1 ppbv PNs and 10 ppbv NO₂) | Numerical simulation | Keehan et al. (2020) |
| RO₂ + NO₂ | 453 K | CEAS | PAN | -ve | Factor 1.43 (3.3 ppbv PAN and 10.5 ppbv NO₂) | Numerical simulation | Li et al. (2021) |
| RO₂ + NO₂ | 653 K | CEAS | PAN | -ve | Factor 1.08 (3.3 ppbv PAN and 10.5 ppbv NO₂) | Numerical simulation | Li et al. (2021) |
| RO₂ + NO₂ | 453 K | CRDS | PAN | -ve | Factor 1.15 (2.8 ppbv PAN and 10 ppbv NO₂) | Glass beads, standard addition experiments | Lin et al. (2024) |
| RO₂ + NO₂ | 633 K | CRDS | PAN | -ve | Factor 1.04 (1.71 ppbv PAN and 10 ppbv NO₂) | Glass beads, standard addition experiments | Lin et al. (2024) |
| RO₂ + NO₂ | 448 K | CRDS | PAN | -ve | Factor 1.61 (2 ppbv PAN and 11 ppbv NO₂) | Numerical simulation | This work |
| RO₂ + NO₂ | 648 K | CRDS | PAN | -ve | Factor 1.08 (2 ppbv PAN and 11 ppbv NO₂) | Numerical simulation | This work |
| RO₂ + NO₂ | 723 | LIF | EN | -ve | Not significant | -- | Di Carlo et al. (2013) |
| RO₂ + NO₂ | 723 K | CRDS | IPN | -ve | Factor 1.05 (12 ppbv NO₂) | Numerical simulation | Thieser et al. (2016) |
| RO₂ + NO₂ | 648 K | CRDS | IPN | -ve | Factor 1.03 (6 ppbv NO₂) | Numerical simulation | Sobanski et al. (2016) |
| RO₂ + NO₂ | 633 K | CRDS | EHN | -ve | Factor 1.27 (2.03 ppbv EHN and 12 ppbv NO₂) | Glass beads, standard addition experiments | Lin et al. (2024) |
| RO₂ + NO₂ | 648 K | CRDS | Limonene-nitrate | -ve | Upper limit: Factor 1.09 (3.37 ppbv Limonene-nitrate and 20 ppbv NO₂) | -- | This work |
| RO₂ + NO₂ | 433 K 633 K | CAPS | PAN IPN | -ve | Not significant | -- | Ohara et al. (2024) |

Notes:

CRDS = cavity ring-down-spectroscopy, CAPS = cavity attenuated phase shift spectroscopy, CEAS = cavity-enhanced absorption spectroscopy, CIMS = chemical ionization mass spectrometry, LIF = laser induced fluorescence, PERCA = peroxy radical amplifier. PAN is $CH_3C(O)OONO_2$, PPN is $C_2H_5C(O)OONO_2$, IPN is isopropyl nitrate $((CH_3)_2CH_2ONO_2,CH_3)$, MN is methyl-nitrate $(CH_3ONO_2)$, EN is ethyl-nitrate $(C_2H_5ONO_2)$, NPN is n-propyl-nitrate $(CH_3CH_2CH_2ONO_2)$, EHN is 2-ethylhexyl-nitrate. a) $S'_{380} = S_{380} / (0.0694 \times \ln(S_{380}) - 0.308) \times (0.0115 \times [O_3] + 0.557)$ where $S_{380}$ is the observed total signal and $S'_{380}$ is the corrected signal. b) $NO_2$ loss $(L_{NO2}$ (ppbv) $= 7.43 \times$

$10^{-4} [NO_2][O_3])$. c) $k_{eff}(473$ K$) = 2.5 \times 10^{-14}$ cm³ molecule⁻¹ s⁻¹, $k_{eff}(723$ K$) = 6.2 \times 10^{-14}$ cm³ molecule⁻¹ s⁻¹ where $k_{eff}$ is an effective rate coefficient for the reaction between $O_3$ and NO.

In this study we assess the bias caused by ambient NO on ∑PNs and ∑ANs measurements, with a focus on biogenically derived ANs formed in the $NO_3$-induced oxidation of isoprene and limonene, which so far have been neglected in the studies reviewed in Tab. 1. We show that the use of unsaturated surrogate alkyl-nitrates (e.g. IPN) to estimate bias in TD-inlets leads to a

different effect compared to alkyl nitrates derived from the oxidation of common biogenic trace gases such as terpenoids and as such is inappropriate for estimating and correcting bias.



## 3 Experimental

### 3.1 TD-Cavity ring-down spectrometer

The measurements were performed using a 5-channel cavity ring-down spectrometer, which has been described in detail

previously (Sobanski et al., 2016; Dewald et al., 2021). In this study, we use only the three cavities that detect $NO_2$. Each cavity consists of a stainless steel tube (70 cm length and inner diameter 8 mm) that is internaly coated with Teflon (Chemours, FEP, TE 9568) to reduce losses of trace-gases during transmission. Perfluoroalkoxy polymer (PFA) T-pieces at both ends of the tubes provide connections to the sample inlet-line (see Fig. 1) and the outlet to the pump. Sample air enters the cavities at a flow rate of 2.1 slpm in each channel. Two highly reflective mirrors (transmission of 20 ppm at 409 nm, FiveNine Optics)

are located behind each PFA T-piece at a distance of 93.5 cm ($d$) apart. The mirrors are purged with dry zero air (CAP 180, Fuhr GmbH) which results in a reduction of the effective optical path length, $l$, compared to the mirror distance. To record a ring-down signal, light emitted by a laser diode is coupled into the cavities using 50 µm core optical fibres and collimators. Using a photomultiplier tube, which converts the transmitted light intensity into an analogue electrical signal, an exponential decay can be observed when the laser is temporarily switched off. The concentration of $NO_2$ in each cavity is calculated from:

$$[NO_2] = \frac{d}{l} \frac{(k - k_0)}{c \cdot \sigma_\lambda},\qquad(1)$$

where $k_0$ and $k$ are the decay rate coefficients without and with absorbing/scattering medium, $c$ is the speed of light and $\sigma_\lambda$ is the effective absorption cross section resulting from the overlap of the laser emission and the $NO_2$ absorption spectrum (Vandaele et al., 1998).

In the original version of this instrument (Sobanski et al., 2016; Dewald et al., 2021), three cavities were operated at 409 nm

to detect either $NO_2$ or $\sum PNs + NO_2$ or $\sum ANs + \sum PNs + NO_2$. We refer to these as the "$NO_2$-cavity", the "PNs-cavity" and the "ANs-cavity", respectively. The $NO_2$-cavity sampled air via a 60 cm long PFA inlet (8 mm inner diameter) at ambient temperature. The PNs-cavity sampled via a 55 cm long PFA tubular (8 mm inner diameter), the first 20 cm of which were heated to 448 K to convert PNs to $NO_2$. The ANs-cavity sampled air via a 55 cm long quartz tube (12 mm inner diameter), where the first 20 cm were heated to 648 K, to convert ANs to $NO_2$.

In this work, we examine the advantages of detecting $NO_X$ rather than $NO_2$ when using a TD-CRDS to measure peroxy nitrates and alkyl nitrates derived from biogenic precursors. The instrument was modified to detect $NO_X$ by adding $O_3$ (Fig. 1) which was generated by passing zero air over a low-pressure Hg-lamp emitting at 185 nm. The $O_3$ was divided equally by critical orifices into three identical reaction volumes made of 88 cm long PFA tubing (0.5 inch outer diameter, residence time of 1.8 s). About 12 ppmv $O_3$ were needed to fully convert NO to $NO_2$. This was determined experimentally by changing the flow

through the critical orifices while monitoring $NO_2$ for a fixed amount of NO. A numerical simulation of the reactions describing the formation and loss of $NO_2$ (Fig. S1) indicated a conversion efficiency of 98.7 %.





## 3.2 PAN sources

Two different sources of PAN were used in this study. In the first, PAN was prepared via the photolysis of acetone in air in the presence of NO as described originally by Warneck and Zerbach (1992) and improved by e.g. Flocke et al. (2005). In this

case, 5 sccm of acetone (4.6 % in synthetic air, Air Liquide) and 10 sccm of NO (1 ppmv, Air Liquide) were mixed with 100 sccm of zero air before flowing into a quartz reactor equipped with a phosphor-coated, low-pressure mercury lamp (Pen-Ray) to photolyse acetone at ~ 285 nm (R16). Based on the volume flow rate and the volume of the reactor (~ 243 cm$^3$), the residence time of the gas flowing through the reactor was calculated to be ~ 120 s. PAN is formed at a yield of ~ 90-95 % (relative to NO consumed) by the sequence of reactions (R16-R19).

155          $CH_3C(O)CH_3 + h\nu$       $\rightarrow$       $CH_3C(O) + CH_3$                                                        (R16)

         $CH_3C(O) + O_2 + M$       $\rightarrow$       $CH_3C(O)O_2 + M$                                                 (R17)

         $CH_3C(O)O_2 + NO$       $\rightarrow$       $CH_3CO_2 + NO_2$                                                 (R18)


         $CH_3C(O)O_2 + NO_2 + M$    $\rightarrow$       $CH_3C(O)O_2NO_2 + M$                                      (R19)

The mixture exiting the photochemical reactor was diluted with 5.2 slpm zero air to provide PAN mixing ratios of 1.7 ppbv. In order to ensure a known concentration of PAN, such photochemical PAN-sources are operated in the NO-limited chemical

regime, i.e. sufficent peroxyl radicals are present to initially convert a NO to $NO_2$ and then to PAN. High radical densities implies that other chemical processes involving peroxyl radicals take place. In section 4.1, we later discuss the concentrations and role of the organic and inorganic photochemical products other than PAN that are generated in the reactor.

Alternatively, PAN was synthesised using wet-chemical methods (Gaffney et al., 1984; Talukdar et al., 1995). Tridecane (10 mL, ≥ 99 %, Sigma-Aldrich) and peroxyacetic acid (PAA) (2.5 mL, 35 % by weight in dilute acetic acid, Acros Organics)

were placed in a 100 mL wide-necked round-bottom flask and cooled to 273 K in an ice-water bath. Cold sulphuric acid (2.5 mL, 98 %, Roth) and then cold nitric acid (1.3 mL, ≥ 65 %, Sigma-Aldrich) were added dropwise while stirring. The contents of the flask were allowed to stand for 25 minutes at 273 K with vigorous stirring and then transferred to a separatory funnel containing ice-cold Milli-Q water (25 mL). The aqueous phase was discarded and the organic phase was shaken four times with 25 mL of cold Milli-Q water each time. The organic phase was then dried for 30 minutes at 273 K with magnesium

sulphate (1 spatula, Alfa Aesar). After filtration, PAN was obtained as a colourless liquid in tridecan and stored at 243 K until use. During experiments, PAN was cooled to 273 K in a cryostat and eluted into the gas-phase by flowing 100 sccm of air over the sample, with further dilution with 5.2 slpm zero air resulting in mixing ratios between 2.0 and 6.5 ppbv.

## 3.3 AN sources

Limonene nitrates and isoprene nitrates were formed by the reaction of limonene or isoprene with $NO_3$ in air in the 1 m$^3$

simulation chamber SCHARK (Simulation Chamber for Atmospheric Reactions and Kinetics), which was  operated at ambient





pressure and temperature (1 atm, 298 K) in dynamic-flow mode (Dewald et al., 2021). Residence times in the chamber were between 57 and 92 minutes depending on the flow rate. Gases entering the chamber passed through a circular loop of PFA tubing (located at the bottom of the chamber) with 10, equally spaced, 1 mm-diameter holes pointing upwards. The velocity of gas at the exit of the holes is estimated at ~2 m s$^{-1}$, which causes rapid turbulent mixing. Mixing is also assisted by a ~5 cm diameter, teflon coated fan located centrally at the bottom of the chamber. Experiments with pulsed addition of $O_3$, showed that the mixing time in the chamber was < 40 s, which is negligible compared to the overal residence time.

Two methods were used to produce $NO_3$. In the first, $NO_3$ was formed by the thermal decomposition of $N_2O_5$ (R7) which was eluted into the chamber by passing zero air (50 - 600 sccm) over $N_2O_5$ crystals held at temperatures between 205 K and 195 K (acetone-dry ice bath). $N_2O_5$ was synthesised by flowing 150 sccm of NO (5% in $N_2$, Westfalen) and 200 sccm of $O_3$ in $O_2$ (~ 10 % $O_3$, generated via electrical discharge in $O_2$) through a ~ 2 L, glass reaction vessel and trapped as colourless crystals at 195 K (acetone-dry ice bath). $N_2O_5$ crystals formed this way were stored at 243 K until use. Alternatively, $NO_3$ was obtained by bubbling 50-200 sccm of zero air through a 0.5 M solution of cerium(IV) ammonium nitrate (CAN in 6 M $HNO_3$) that was irradiated at 365 nm (Lambe et al., 2023). The flow containing the $NO_3$ was diluted by 15 slpm zero air before entering the chamber. The thermal decomposition of $N_2O_5$ has the advantage that the lifetime of $N_2O_5$ is sufficiently long (~23 s at 298 K) to ensure that $NO_3$ is formed reasonably homogeneously throughout the chamber. The disadvantage of using $N_2O_5$ is the release of $NO_2$ during its thermal decomposition. The photochemical CAN source suffers from the disadvantage that the $NO_3$ (added via a point source) may react before mixing is complete. i.e. the lifetime of $NO_3$ in the presence of (typically) 30 ppbv of limonene or isoprene is only 0.1 -2 s, which is shorter than the mixing time. Isoprene and limonene were introduced into the chamber via flow-controllers attached to steel canisters containing isoprene (98%, Acros Organics, 45 ppmv in He) or limonene (96%, Sigma-Aldrich, 44 ppmv in He) in He (5.0, Westfalen), resulting in ANs mixing ratios between 0.47 and 4.4 ppbv.

The nitrates generated in this manner from isoprene are a mixture of C5-nitrooxyhydroperoxides, C5-nitrooxycarbonyl, C5-hydroxynitrate and also C10-nitrooxyperoxide (ROOR), with the relative concentrations depending on the fate of the initially formed nitrooxyperoxy radicals (IUPAC 2024). The limonene nitrates formed are mainly C10-hydroxynitrates at yields of ~ 20 % (Spittler et al., 2006).

## 3.4 Experimental procedure

In order to investigate the influence of NO and $NO_2$ on the detection of PAN and biogenically derived ANs via TD-CRDS, different amounts of NO (0 – 22.5 ppbv) and $NO_2$ (0 – 20 ppbv) were added to the TD-inlet of the CRDS along with various amounts of PAN, isoprene-derived ANs and limonene-derived ANs. These experiments were carried out with the cavities operating both in "$NO_2$-mode" and in "$NO_X$-mode" by adding $O_3$ (see section 3.2).



# 4 Results and discussion

## 4.1 The influence of NO on the measurement of PNs (Photochemical PAN source, PAN measured as NO₂)

The presence of NO during the measurement of PNs using TD-inlets leads to a positive bias in ambient PNs concentration, which is related to the oxidation of NO to $NO_2$ via reactions with peroxyl radicals (see section 2) (Sobanski et al., 2016; Thieser et al., 2016). To re-investigate this measurement artefact, extend previous studies from this group and provide a solution to improve the reliability of TD-CRDS instruments designed to measure ΣPNs, we have performed experiments using both a photochemical source and a diffusion PAN source (see section 3.2). Fig. 2 summarises the results of an experiment in which NO was added to photochemically produced PAN ($\approx$ 1.7 ppbv). In these experiments, $O_3$ was NOT added to the tubing in front of the cavities (low-pressure Hg-lamp off) so that the cavities detected $NO_2$ (and not $NO_X$). The mixing ratios of $NO_2$ measured in the cavities with inlet temperatures of 448 and 648 K are plotted against the amount of NO added, which was calculated from its partial flow and mixing ratio in the bottle (1 ppmv in $N_2$). For both datasets, the $NO_2$ measured in the room temperature cavity (present as an impurity in the PAN sample) has already been subtracted.

Ideally, only one molecule of $NO_2$ would be generated for each PAN molecule thermally dissociated and the data would describe a horizontal line at ~1.7 ppbv. As Fig. 2a shows this is clearly not the case. In both cavities with heated inlets the addition of NO leads to the formation of additional $NO_2$. This is related to co-formation of the acetyl peroxy radical from PAN in the heated inlet at 448 K (R20) which reacts with NO (R21) to form $NO_2$ and (in air) a methyl peroxy radical (R21a-b). The methyl peroxy radical can react with NO to form a further $NO_2$ and $HO_2$ (R22); the latter can form a third $NO_2$ in the reaction between $HO_2$ and NO (R23). The OH radical formed in this process reacts either with NO to form HONO or with $NO_2$ to form $HNO_3$ both of which resist dissociation in the 448 K TD-inlet (Thieser et al., 2016; Friedrich et al., 2020).

| | | | |
|---|---|---|---|
| $CH_3C(O)O_2NO_2 + M$ | $\rightarrow$ | $CH_3C(O)O_2 + NO_2 + M$ | (R20) |
| $CH_3C(O)O_2 + NO$ | $\rightarrow$ | $CH_3CO_2 + NO_2$ | (R21) |
| $CH_3CO_2 + M$ | $\rightarrow$ | $CH_3 + CO_2 + M$ | (R21a) |
| $CH_3 + O_2 + M$ | $\rightarrow$ | $CH_3O_2 + M$ | (R21b) |
| $CH_3O_2 + NO + (O_2)$ | $\rightarrow$ | $HCHO + HO_2 + NO_2$ | (R22) |
| $HO_2 + NO$ | $\rightarrow$ | $OH + NO_2$ | (R23) |

Additionally, the radicals involved (e.g. $RO_2$, RO, OH) can be lost to the walls of the TD-inlet, which modifies the extent of the conversion of NO to $NO_2$. In our experiments with a TD-inlet at 448 K and 1.7 ppbv PAN, the addition of 20 ppbv NO results in detection of $> 5$ ppbv $NO_2$. At this temperature, the increase in $NO_2$ per NO added is lower at high NO than at low NO (i.e. the curve flattens at high NO).

At 648 K the chemistry is different as the acetyl-peroxy radical decomposes into an acetyl radical and oxygen (R24) or isomerises to $CH_2C(O)OOH$ (R25) (Lee et al., 2002; Carr et al., 2011; Thieser et al., 2016) which decomposes thermally to OH and a singlet α-lactone (R26) (Carr et al., 2011; Thieser et al., 2016). The acetyl radical from equation R24 can decompose to $CH_3$ and CO (R27), react with $O_2$ to form an acetyl peroxy radical (R28) or form OH and the singlet α-lactone (R29) (Tyndall



et al., 1995; Carr et al., 2007; Chen and Lee, 2010; Carr et al., 2011; Groß et al., 2014; Papadimitriou et al., 2015; Thieser et al., 2016). The methyl radical formed in equation (R27) reacts with oxygen to form a methyl peroxy radical (R21b), which

reacts with NO to form $NO_2$ (R22).

$$CH_3C(O)O_2 + M \quad \rightarrow \quad CH_3C(O) + O_2 + M \qquad (R24)$$

$$CH_3C(O)O_2 + M \quad \rightarrow \quad CH_2C(O)OOH + M \qquad (R25)$$

$$CH_2C(O)OOH \quad \rightarrow \quad {}^1C_2H_2O_2 + OH \qquad (R26)$$

$$CH_3C(O) \quad \rightarrow \quad CH_3 + CO \qquad (R27)$$

$$CH_3C(O) + O_2 + M \quad \rightarrow \quad CH_3C(O)O_2 + M \qquad (R28)$$

$$CH_3C(O) + O_2 \quad \rightarrow \quad {}^1C_2H_2O_2 + OH \qquad (R29)$$

For the 648 K inlet the bias is even larger (a factor >5 increase in $NO_2$ going from zero to 20 ppbv NO) and appears to flatten-off more slowly. This result was unexpected as previous studies from this laboratory have indicated that at high temperatures, the bias is weaker owing to thermal decomposition of $CH_3C(O)O_2$.

In order to explore this effect, a thermogram of the photochemical PAN source was measured while continuously adding 13 ppbv of NO to the TD-inlet. The result is displayed in Fig. 3, which confirms that NO to $NO_2$ conversion continues past the PAN-maximum close to 440 K with a doubling of the $NO_2$ mixing ratio at 648 K.

In order to understand the origin of the additional $NO_2$ signal at high temperatures, a further experiment was carried out in which the photochemical PAN source was switched on, but without adding NO to produce PAN. This can be seen in Fig. 2b

and shows that the conversion of NO to $NO_2$ in the 648 K cavity continues despite the absence of PAN and thus cannot be explained solely by reactions initiated by formation of the acetyl peroxy radical. A smaller conversion of NO to $NO_2$ is also observed in the 448 K cavity in the absence of PAN.

In the absence of NO, peroxy radicals formed by acetone photolysis in the photochemical reactor undergo a series of self- and cross-reactions that lead to a number of organic end-products that can be transported from the photochemical reactor to the

TD-inlets. Note that within the photochemical reactor significant concentrations of hydroxy- and peroxy-radicals are only present in the steady-state when irradiated and do not survive transport through the tubing ($\approx$ 2 m of ¼ inch PFA) to the TD-inlets.

The two initially formed peroxyl radicals (R16-17 and R21b) $CH_3C(O)O_2$ and $CH_3O_2$ can undergo self- and cross-reactions in the photochemical reactor, a sub-set of which are shown below:

$$CH_3C(O)O_2 + CH_3O_2 \quad \rightarrow \quad CH_3CO_2 + CH_3O + O_2 \qquad (R30)$$

$$CH_3C(O)O_2 + CH_3C(O)O_2 \quad \rightarrow \quad 2\ CH_3CO_2 + O_2 \qquad (R31)$$

$$CH_3O_2 + CH_3O_2 \quad \rightarrow \quad 2\ CH_3O + O_2 \qquad (R32)$$

$$CH_3O + O_2 \quad \rightarrow \quad HO_2 + HCHO \qquad (R33)$$

$$HO_2 + HO_2 \quad \rightarrow \quad H_2O_2 + O_2 \qquad (R34).$$



The $CH_3CO_2$ thermally decompose in air to form $CH_3$ and $CO_2$ (R21a) so that $CH_3C(O)O_2$, $CH_3O_2$ and $HO_2$ are initially present. $HO_2$, formed in R33 reacts with all organic peroxy radicals in both radical termination and radical propagation steps:

$$CH_3C(O)O_2 + HO_2 \qquad \rightarrow \qquad CH_3C(O)OOH + O_2 \qquad\qquad (R35)$$

$$\rightarrow \qquad CH_3C(O)OH + O_3 \qquad\qquad (R36)$$

$$\rightarrow \qquad OH + CH_3 + O_2 + CO_2 \qquad\qquad (R37)$$

$$CH_3O_2 + HO_2 \qquad \rightarrow \qquad CH_3OOH + O_2 \qquad\qquad (R38).$$

The OH formed in reaction R37 will react with acetone (which is present in large excess) to form another peroxy radcial:

$$OH + CH_3C(O)CH_3 + O_2 \qquad \rightarrow \qquad CH_3C(O)CH_2O_2 + H_2O \qquad\qquad (R39).$$

The acetonyl peroxy radical ($CH_3C(O)CH_2O_2$) will also undergo reaction with other $RO_2$, eventually forming peroxides and C1 and C2-peroxyl radicals. In summary, the chemistry taking place in the PAN-photochemical source generates a large variety

of oxidised organic trace gases (not only PAN) that may enter the TD-inlet at 648 K. Of particular interest with respect to explaining the data of Fig. 2 are the thermally unstable peroxides such as $CH_3C(O)OOH$, $CH_3OOH$ and $H_2O_2$. These trace gases are stable at room temperature but have lifetimes with respect to thermal dissociation to OH of 0.06 s (Sahetchian et al., 1992), 0.6 s (Kirk, 1965) and 41 s (Yang et al., 2021) at 648 K, respectively.

The thermal dissociation of $CH_3C(O)OOH$ in the TD-inlet leads (via the thermal dissociation of initially formed $CH_3CO_2$

(R21)) to the formation of an OH radical and $CH_3O_2$ (R40) (Schmidt and Sehon, 1963; Atkinson et al., 2006):

$$CH_3C(O)OOH\ (+ O_2) \qquad \rightarrow \qquad CH_3O_2 + CO_2 + OH \qquad\qquad (R40).$$

The $CH_3O_2$ thus formed can, via reactions R22 and R23, account for some of the conversion of NO to $NO_2$ observed. However, both R23 and R40 generate OH radicals, which can react with acetone in the TD-inlet. The rate coefficient for OH + acetone at 648 K is $1.4 \times 10^{-12}$ $cm^3$ molecule$^{-1}$ s$^{-1}$ (Vasudevan et al., 2005) which results in an OH-lifetime in the heated inlet of ~ 1 ms

with respect to reaction with acetone. This results in the formation of the acetonyl peroxy radical (R39) which then also reacts with NO (R41) to form other peroxyl radicals ($HO_2$, $CH_3C(O)O_2$ and $CH_3O_2$ via reactions R42 and R43) that can also convert NO to $NO_2$.

$$CH_3C(O)CH_2O_2\ + NO \qquad \rightarrow \qquad CH_3C(O)CH_2O + NO_2 \qquad\qquad (R41)$$

$$CH_3C(O)CH_2O + O_2 \qquad \rightarrow \qquad CH_3C(O)CHO + HO_2 \qquad\qquad (R42)$$

$$CH_3C(O)CH_2O + \Delta H\ (+ O_2) \qquad \rightarrow \qquad CH_3C(O)O_2 + HCHO \qquad\qquad (R43)$$

The hypothesised role of the thermal dissociation of $CH_3C(O)OOH$ and $H_2O_2$ at 648 K as source of peroxyl radicals in the TD-inlet was tested by flowing $CH_3C(O)OOH$ (~ 1 ppbv) or $H_2O_2$ (2 – 10 ppmv, calculated from vapour pressure) sample through the TD-inlet in the presence of various amounts of NO (Fig. 4). Note that the $NO_2$ measured in the room temperature cavity

(present as an impurity of 3.5 % in the NO bottle) has been subtracted from the data obtained in the 648 K TD-inlet. As can be seen in Fig. 4a, the addition of up to 17.5 ppbv NO to 1 ppbv PAA or 2-10 ppmv $H_2O_2$ results in 0.6 ppbv or 5 ppbv $NO_2$, respectively, confirming that the thermal decomposition of PAA and $H_2O_2$ at 648 K is a source of peroxyl radicals. The



hypothesis was further tested by varying the TD-inlet temperature when flowing either PAA or $H_2O_2$ in the presence of 12 ppbv NO. As shown in Fig. 4b, in both cases we observed efficient conversion of NO to $NO_2$ at temperatures above 500 K for

PAA and above 400 K for $H_2O_2$. In separate experiments, we observed that the thermal dissociation of acetone also accounts for a small fraction of the overestimation $NO_2$ formed (400 pptv $NO_2$ at 43 ppmv acetone and 16 ppbv NO), which is confirmed by its thermogram (Fig. S2).

In order to gain insight into the identity and relative concentrations of the trace gases exiting the photochemical reactor, two sets of numerical simulations were carried out with FACSIMILE (Curtis and Sweetenham, 1987), both with and without the

addition of NO to the photochemical PAN source (i.e. with and without PAN formation, Tab. S1). The results are summarised in Fig. S3 which displays the evolution of products in the photochemical reactor as a function of time. At 89 ppbv NO (Fig S3a) 88 ppbv PAN is produced. Note that, following dilution, this would result in ~ 2 ppbv of acetone in the TD-inlets as observed experimentally. The simulation indicates ppmv mixing ratios of several organic traces gases in the photochemical reactor with 570 ppmv HCHO, 60 ppmv $CH_3OOH$ and 1 ppmv $H_2O_2$ after 120 s reaction time. In the absence of NO (and thus

PAN, Fig. S3b) higher mixing ratios of these organics are generated, as expected. In this case the simulation predicts 1650 ppmv HCHO, 125 ppmv $CH_3OOH$, 37 ppmv $CH_3C(O)CH_2O_2$, 15 ppmv PAA and 2 ppmv $H_2O_2$.

In summary, the photochemical PAN source is not well suited for characterising the inlet chemistry for alkyl nitrate detection via thermal decomposition to $NO_2$ as it results in significant NO to $NO_2$ conversion owing to the presence of thermally labile trace gases such as peracetic acid and hydrogen peroxide. The presence of organic acids and their possible interference in

chemical-ionisation mass spectrometric studies of PAN has been documentde previously (Mielke and Osthoff, 2012).

## 4.2 The influence of NO and $NO_2$ on the measurement of PNs (PAN diffusion source)

The results presented above clearly demonstrate that the photochemical PAN source generates not only PAN but a number of oxidised organics that can influence the chemistry in the TD-inlet. This precludes an accurate assessment of the bias caused by the presence of NO or $NO_2$ when attempting to measure ΣPNs in real air masses. For this reason, further experiments were

performed with a PAN diffusion source which is not expected to have any significant impurities. The IR spectrum (Fig. S4) of a gas sample eluted from the PAN diffusion source is in good agreement with the literature and does not reveal the presence of high levels of impurities.

Figure 5 shows the results of three experiments in which PAN was added to zero air at mixing ratios of 2.0 and 6.5 ppbv and detected (following thermal dissociation) in both the 448 K and 648 K inlets as $NO_2$. As described in the experiments above,

the addition of various amounts of NO to the inlet results in a strong positive bias to the measurement, with a factor ~2 more $NO_2$ measured at 15 ppbv NO. Compared to the photochemical PAN source the bias ($NO_2$ formed per ppbv NO added) is smaller, which is related to the absence of impurities such as $CH_3C(O)OOH$. The solid lines in these plots are the results of numerical simulations that consider the formation and loss of radicals (both gas-phase and heterogeneous) in the TD-inlets and the resultant oxidation of NO to $NO_2$. The simulations are described in detail in previous publications from this group (Sobanski

et al., 2016; Thieser et al., 2016). While the simulations reproduce the bias caused by NO oxidation to $NO_2$ reasonably well in




these laboratory experiments with only one major trace-gas present (PAN) apart from NO, the situation in the real atmosphere will be more complex with potentially several different PAN species and many other trace gases that also interact with OH in the TD-inlets to form other peroxyl radicals that may react with NO.

An effective approach to remove the bias caused by NO oxidation to $NO_2$ has been described for a TD-inlet system in this

group, designed to detect $NO_y$ via conversion to $NO_2$ (Friedrich et al., 2020). In this case, the air exiting the TD-inlet is allowed to cool before adding sufficient $O_3$ to convert NO to $NO_2$ (see section 3.1) and PAN is measured as $NO_X$ rather than $NO_2$. The results of an experiment in which PAN was detected as $NO_X$ is displayed as the red and blue lines in Fig. 5. In this case, ~3.7 ppbv PAN were initially present and up to 17 ppbv of NO were added. The clear dependence of the retrieved PAN mixing ratio on the NO mixing ratio is no longer apparent when using either the 448 K or 648 K TD-inlet and alleviates the need to

perform numerical simulations to correct the data.

The bias caused by the recombination of $NO_2$ with $CH_3C(O)O_2$ (section 2) was investigated when both $NO_2$ and $NO_X$ were detected. The results of two experiments in which either 2.0 or 6.5 ppbv PAN from the diffusion source were sampled into the TD-CRDS with the addition of various amount of $NO_2$ are shown in Fig. 6. The effect of the recombination of the acetyl peroxy radical with $NO_2$ at 448 K can be seen in the data. Initial mixing ratios of PAN of 2.0 ppbv and 6.5 ppbv are only

detected as 1.1 ppbv and 3.4 ppbv $NO_2$, respectively. At 648 K the effect is significantly smaller than at 448 K, reflecting the thermal instability of $CH_3C(O)O_2$ at the higher temperature. The solid black and grey lines in this plot are the results of numerical simulations (Sobanski et al., 2016; Thieser et al., 2016). For the detection of $NO_X$ (blue and red symbols in Fig. 6) a similar effect was observed as for the detection of $NO_2$. Thus, numerical simulations are still required to correct for the bias caused by the oxidation of $NO_2$. Within the experimental uncertainty, there is no significant difference in the size of the

negative bias when detecting PAN as either $NO_2$ or $NO_X$. This is readily understandable, as the recombination of $CH_3C(O)O_2$ with $NO_2$ to reform PAN after the gas flow cools down (after the heated inlets) is not influenced by the presence of $O_3$ which is added later.

## 4.3 Influence of NO on the measurement of ∑ANs

As described in section 2, a saturated, C3-alkyl-nitrate (IPN) has been widely used to characterise instruments using thermal

dissociation to convert alkyl nitrates to (detectable) $NO_2$. In previous work (Dewald et al., 2021), we have shown how IPN differs in its TD-inlet chemistry from biogenically derived, unsaturated nitrates which undergo surface catalysed reactions (e.g. with ozone) on hot quartz surfaces at 448 K, resulting in the unwanted detection of ANs in the PNs channel and a bias to measurements obtained in environments dominated by biogenic nitrates. Dewald et al. (2021), were able to remove this effect in their measurements of ∑PNs by using a 448 K inlet made of PFA rather than quartz. The instrument still however suffered

from the positive bias caused by reactions of peroxyl radicals with NO, which was enhanced compared to a quartz inlet, as the $RO_2$ radical losses to the PFA-tubing were reduced. Correction of the data obtained with that configuration thus required numerical simulations, which were based on laboratory experiments using IPN (Sobanski et al., 2016; Thieser et al., 2016;





Ohara et al., 2024). We have extended these experiments by using atmospherically more relevant (unsaturated) alkyl nitrates formed by the $NO_3$-initiated degradation of isoprene and limonene.

The results of experiments in which either 0.47 or 4.4 ppbv isoprene nitrate (derived by mixing $N_2O_5$ with isoprene in a reaction chamber, see section 3.3) was sampled into the TD-CRDS with the addition of various amounts of NO are illustrated in Fig. 7a. In these experiments, either $NO_X$ (blue symbols) or $NO_2$ (i.e. the $O_3$ source was switched off, orange symbols) was measured. For the "$NO_2$-mode" experiments $NO_2$ was present at $(14.8 \pm 0.2)$ ppbv which is a consequence of using $N_2O_5$ in the chamber to generate $NO_3$.

In the "$NO_2$-mode" experiment using isoprene derived nitrate, the addition of ~ 22 ppbv of NO results in a doubling of the $NO_2$ signal (factor 2.3). This contrasts the results of Thieser et al. (2016), who, using IPN as surrogate alkyl-nitrate observed larger effects as represented by the black line in Fig. 7a (Thieser et al., 2016).

For limonene derived nitrate(s), the bias when measuring in "$NO_2$-mode" is very large with the addition of the same amount of NO (22 ppbv) resulting in a 690 % overestimation of LIMO-NIT (in the presence of $(19.3 \pm 0.2)$ ppbv $NO_2$). This contrasts

previous results from this group obtained using IPN as surrogate alkyl-nitrate and observed much smaller effects which are represented by the black line in Fig. 7b (Thieser et al., 2016). In summary, at a NO mixing ratio of 20 ppbv (and $NO_2$ ~ 20 ppbv), the fractional conversion of NO to $NO_2$ per 1 ppb of AN present in the sample flow is 7.0 % for IPN, 5.1 % for ISOP-NIT and 25 % for LIMO-NIT. Clearly, the conversion of NO to $NO_2$ in the heated TD-inlet is not only dependent on the amount of NO available to react but also on the nature of the alkyl nitrate. This result is partially intuitive, as the secondary

chemistry resulting from degradation of the longer chain alkyl nitrates will generate more organic peroxy radicals (to oxidize NO to $NO_2$) than IPN. In order to gain some insight into the secondary chemistry involved, numerical simulations of the laboratory data were conducted. The starting point was the reaction scheme used to correct the PNs data as described previously. Rather than attempt to simulate all of the processes occuring at 648 K in the TD-inlet (for which rate coefficients are not available) we varied the number of peroxy radicals generated from the decomposition of the alkoxy radicals formed

in the first reaction of the biogenic derived $RO_2$ with NO and allowed them to react with NO at a rate coefficient of $2.55 \times 10^{-12} \times \exp(380/T)$ cm$^3$ molecule$^{-1}$ s$^{-1}$).

$$\text{LIMO/ISOP-NIT} + \Delta H \quad \rightarrow \quad NO_2 + RO \qquad \text{(R44)}$$
$$RO + O_2 \quad \rightarrow \quad n\,RO_2 \qquad \text{(R45)}$$
$$RO_2 + NO \quad \rightarrow \quad RO + NO_2 \qquad \text{(R46)}$$

To approximately simulate the data, a value of n = 1 was sufficient for isoprene nitrate, whereas a value of 5 was required to reproduce the very large effect observed for limonene nitrate. This is related to the different number of carbon atoms and chemical structure and the different chemical reactions in the presence of NO in the TD-inlet. From our results, it is apparent, that 1) IPN is not a suitable surrogate molecule for the correction of $\sum$ANs, as has previously been practised and 2) given that ambient measurements will normally mixture of many alkyl nitrates, corrections based on numerical simulations that have

been optimised for IPN will not provide reliable data unless NO levels are very low.




Experiments in "NO$_X$-mode" (Fig. 7, blue symbols) in which 22.5 ppbv NO was added to either limonene nitrates (Fig. 7b) or isoprene nitrate (Fig. 7a, both formed by the reaction of NO$_3$ and the corresponding terpene) in the simulation chamber were also conducted and the results are strikingly different to those when NO$_2$ was detected. As expected, there is no significant dependence on NO added between zero and 22 ppbv.

## 5 Conclusions

We have shown that the use of a conventional, photochemical (acetone based) source of PAN can result in incorrect chacterisation of TD-inlets when used to detect PANs and ANs. This is caused by the presence of thermally labile, peroxidic trace gases, which lead to a strong positive bias of NO to NO$_2$ conversion in the ANs inlet. The use of "pure" PAN samples is considered to be superior as it results in more accurate themograms and NO dependencies in the heated inlet designed to detect ANs.

For the first time, we have investigated the potential bias from secondary chemistry (e.g. oxidation of ambient NO to NO$_2$) when using biogenic nitrates derived from terpenoids. The results indicate a larger bias for limonene nitrates and a smaller bias for isoprene nitrates than obtained using a saturated organic nitrate such as IPN. While IPN has the undoubted advantage of being commercialy available and easy to handle, its use to characterise TD-inlets to detect ANs will not lead to results that can be transferred to other (longer chain) nitrates as has been frequently performed. The effect is partially understood in terms of the sequential breakdown of the C10 backbone (e.g. for terpenes) to peroxy radicals that convert NO to NO$_2$.

Inspired by the success of our instrument to detect inorganic and organic NO$_Y$ species as NO$_x$ subsequent to thermal dissociation (Friedrich et al., 2020) we have modified our PANs / ANs instrument in a similar manner (i.e. by detecting NO$_X$ rather than NO$_2$). The results indicate that, when sampling the real atmosphere with NO present, this mode of operation is to be preferred over NO$_2$ detection as the positive bias caused when RO$_2$ formed e.g. from biogenic precursors reacts with ambient NO is eliminated.

In the instrument configuration described here with the PANs TD-inlet made of PFA, the ANs TD-inlet made of quartz and the addition of O$_3$ downstream of the TD-inlets to convert NO to NO$_2$ corrections for the oxidation of NO by O$_3$ and RO$_2$ is no longer necessary and channel cross-talk (i.e. the detection of a fraction of ANs at the PANs inlet) is avoided. The only requirement for simulation is to remove the (negative) bias caused by the recombination of acyl peroxy radicals with NO$_2$ downstream of the TD-inlets. As the dominant atmospheric acyl peroxy nitrate is PAN itself, for which detailed experiments exist to characterise the system, this does not introduce significant uncertainty into the measurement of ΣPNs unless NO$_2$ levels are very large.



## 6 Author contribution

LW and PD performed the experiments, GT assisted with development and deployment of $NO_3$ sources and data acquisition. LW evaluated the data and wrote the first draft of the manuscript. JC conceptualised the SCHARK experiments and with JL contributed to the manuscript. This study was carried out in part fulfilment of the PhD of LW at the Johannes Gutenberg University Mainz.

## 7 Competing Interests

The authors declare that they have no conflict of interest.

## 8 Acknowledgements

We thank Chemours for provision of a FEP sample used to coat the cavities.

PD gratefully acknowledges the Deutsche Forschungsgemeinschaft (project "MONOTONS", project number: 522970430).

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

**10 Figures**

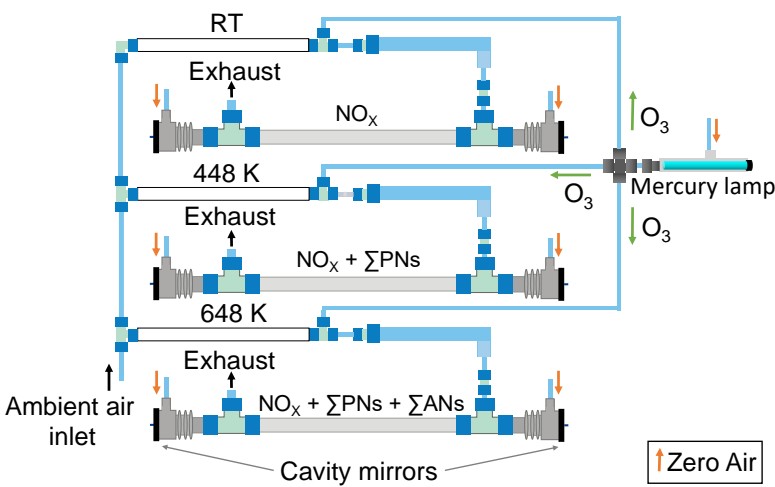

**Figure 1:** Schematic representation of the 409 nm cavities of the 5-channel TD-CRDS for measuring mixing ratios of NO$_X$, NO$_X$ + ∑PNs and NO$_X$ + ∑PNS + ∑ANs (Hg-lamp on). Alternatively, when the Hg-lamp is off NO$_2$ is measured instead of NO$_X$. The coloured arrows 615    represent the gas flows. The temperatures of the different TD-inlets (measured at the outside surface) are shown for each channel.



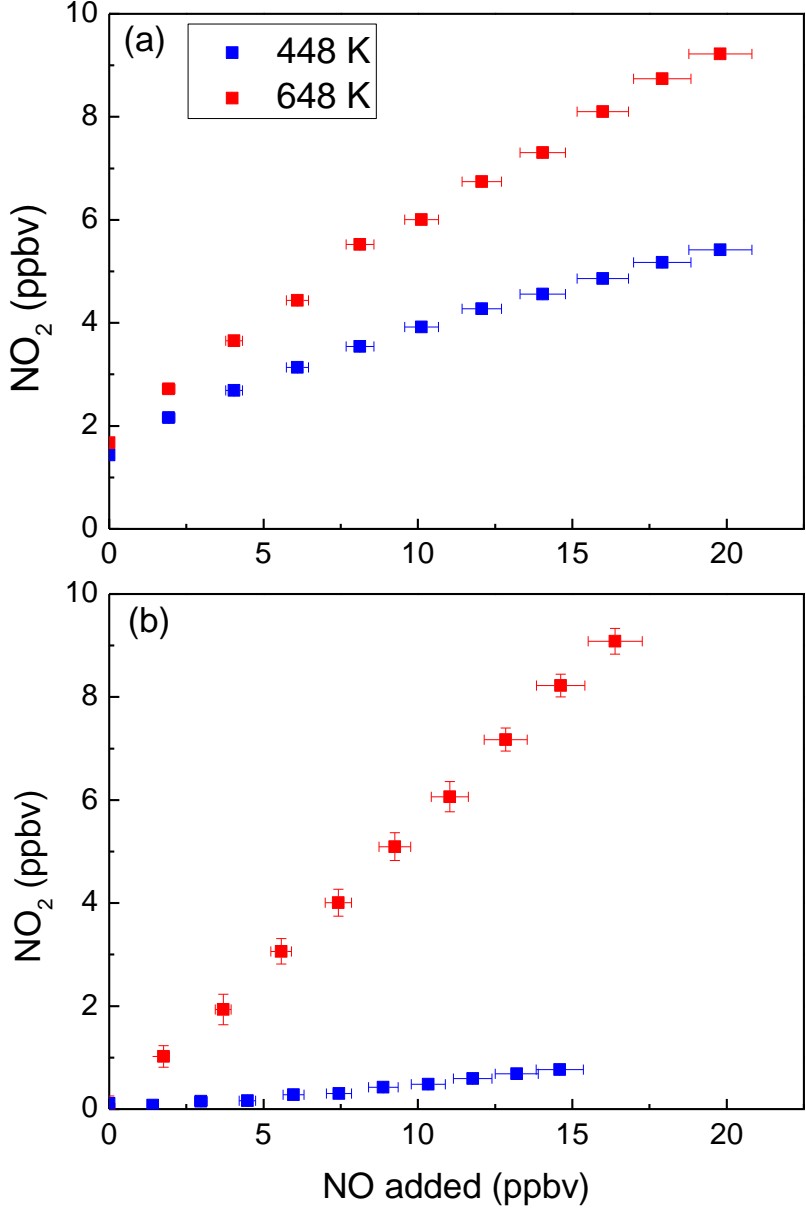

**Figure 2: (a)** NO₂ measured in the cavities with inlet temperatures of 448 K and 648 K while adding NO to photochemically produced PAN. The error bars in the x-direction include uncertainty in the mass flow controllers (1 % of the maximum flow of the MFC) and the uncertainty in the mixing ratio of the NO bottle (5 %). The error bars in the y-direction include the standard deviations (1 σ) of the NO₂ mixing ratios both at 648 K and at room temperature. **(b)** Same as panel (a) but without NO flow into the photochemical PAN source (i.e. no PAN formation).



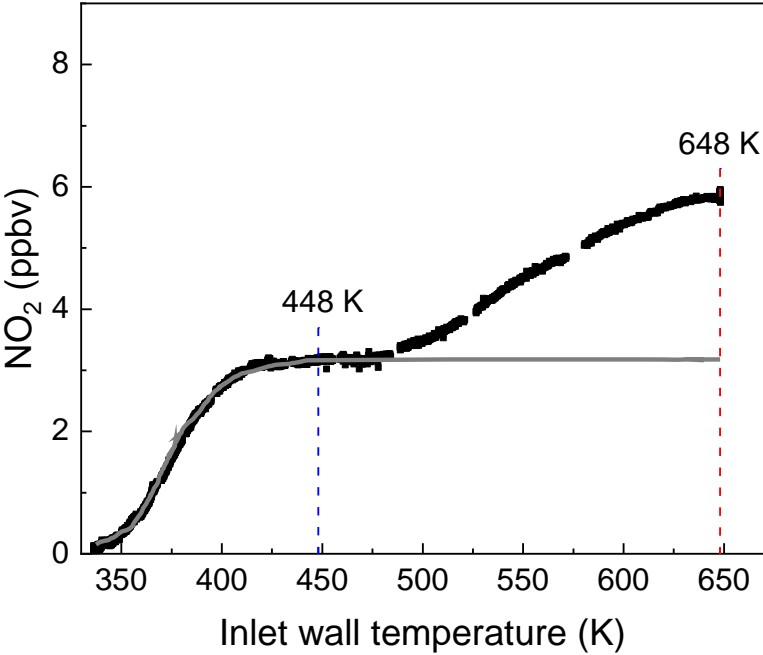

**Figure 3:** Thermogram obtained by adding 13 ppbv NO to 1 ppbv PAN from the photochemical PAN source. The vertical red and blue lines
indicate the normal operating temperatures of the two heated inlets associated with the $\sum$ANs and $\sum$PNs cavities, respectively. Note that the
$NO_2$ measured in the room temperature cavity (impurity present in the NO and PAN flows) has already been subtracted. The grey line
represents the expected thermogram for PAN (i.e. no increase after full dissociation at ~450 K).





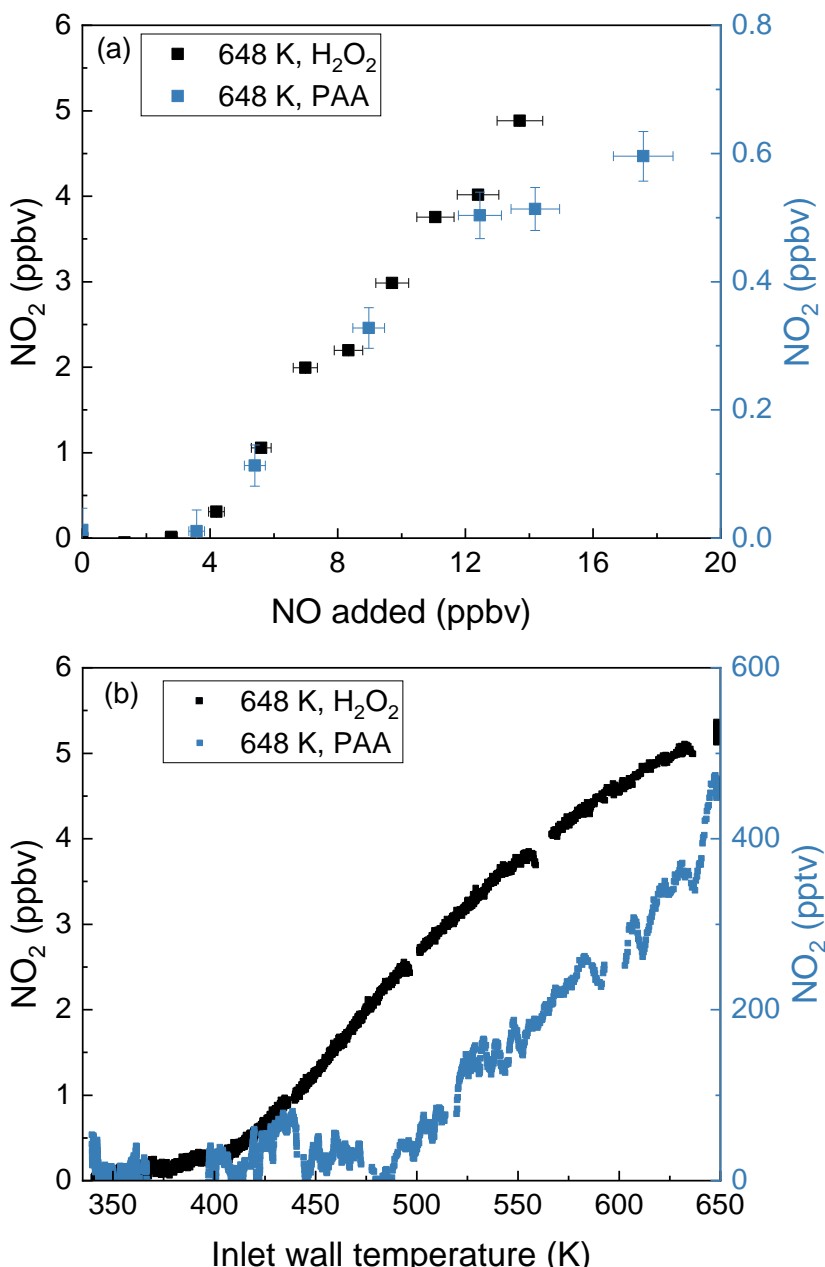

**Figure 4: (a)** NO$_2$ measured in the cavity with an inlet temperature at 648 K while adding NO to PAA ($\approx$ 1 ppbv) and H$_2$O$_2$ (2 – 10 ppmv). The error bars in the x-direction accounts for uncertainty in the mass flows (1 % of the maximum flow of the MFC) and the uncertainty of the mixing ratio of the NO bottle (5 %). The error bars in the y-direction account for uncertainty (1 $\sigma$) in the NO$_2$ mixing ratios both at 648 K and at room temperature. **(b)** Thermograms measured in the ANs-cavity while adding 12 ppbv NO to PAA ($\approx$ 0.3 ppbv) and H$_2$O$_2$ (2 – 10 ppmv).



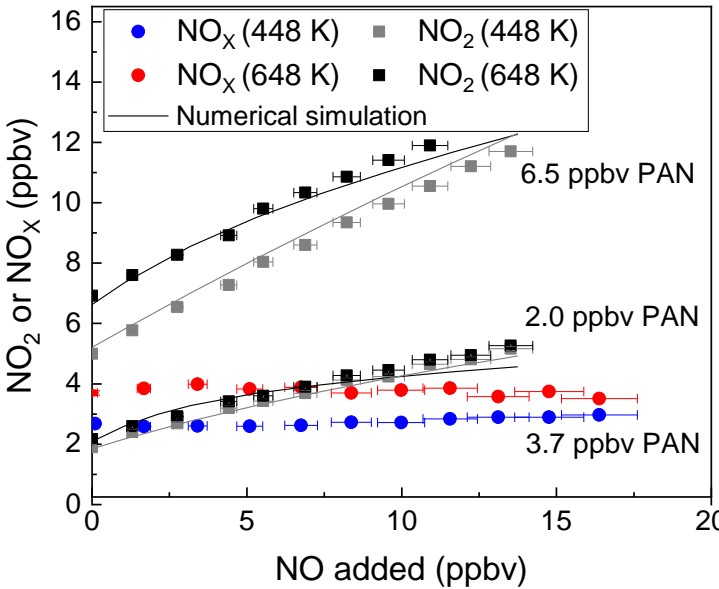

**Figure 5:** Measured mixing ratios of $NO_2$ and $NO_X$ during the addition of NO to PAN from the diffusion source for the cavities with inlet temperatures of 448 K and 648 K. The error bars in the x-direction consider uncertainty in the mass flows (1 % of the maximum flow of the MFC) and uncertainty in the mixing ratio of the NO bottle (5 %). The error bars in the y-direction for 448 K consider the standard deviations (1 σ) of the $NO_2$ mixing ratios both at 448 K and at room temperature. The error bars in the y-direction for 648 K account for the standard deviations (1 σ) of the $NO_2$ mixing ratios both at 648 K and at room temperature The black and grey solid lines show the numerical simulation.





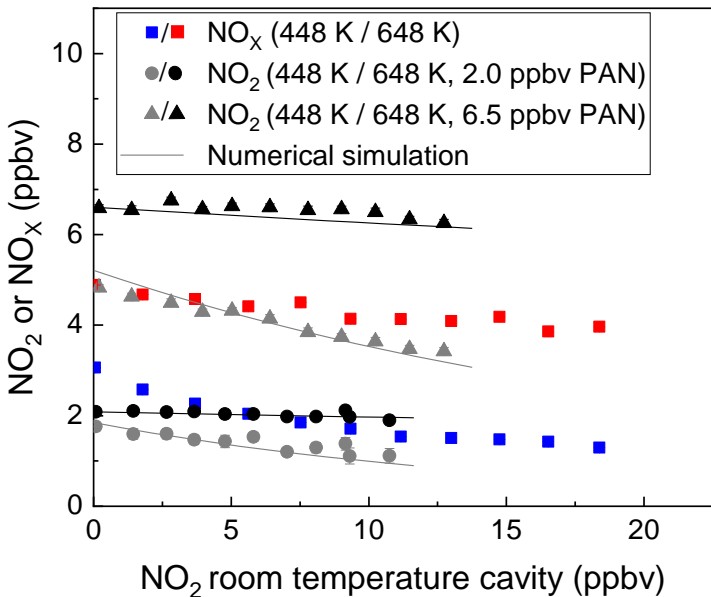

**Figure 6:** Measured mixing ratios of $NO_X$ and $NO_2$ during the addition of $NO_2$ (measured with the cavity with inlet at room temperature) to PAN from the diffusion source. The inlet temperatures were 448 K and 648 K. The error bars in the x-direction represent standard deviation (1 σ) of the $NO_2$ mixing ratio at room temperature, those in the y-direction for 648 K account for uncertainty (1 σ) in the $NO_2$ mixing ratios both at 648 K and at room temperature. The error bars in the y-direction for 448 K account for the standard deviations (1 σ) of the $NO_2$ mixing ratios both at 448 K and at room temperature. The black and grey solid lines are the results of numerical simulations.



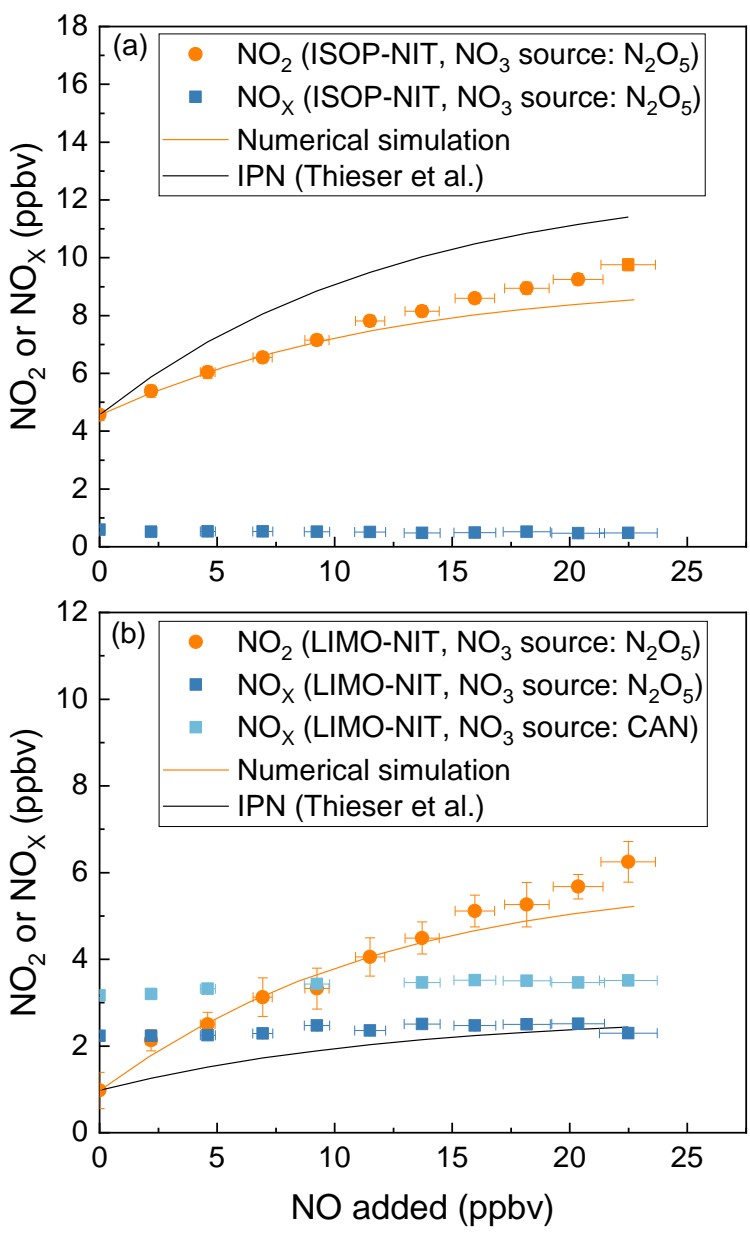

**Figure 7:** Addition of NO to flowing samples of either **(a)** isoprene derived nitrates (ISOP-NIT) or **(b)** limonene derived nitrates (LIMO-NIT). NO$_3$ was either generated with an N$_2$O$_5$ diffusion source or via cerium(IV) ammonium nitrate photolysis (CAN). The error bars in the x-direction consider uncertainty in the mass flows (1 % of the maximum flow of the MFC) and uncertainty in the mixing ratio of the NO bottle (5 %). The error bars in the y-direction consider the standard deviations (1 σ) of the NO$_2$ mixing ratios both at 648 K and at room temperature. The orange lines are the results of numerical simulations of the data at 648 K. The black lines are calculated from the expression: NO$_2$ = 1 + 1.8(exp(-0.08 · NO(ppbv)) (Thieser et al., 2016) as derived for addition of NO to IPN. The NO$_2$ or NO$_X$ measured in the room temperature cavity has been subtracted from that measured in the cavity associated with the 648 K TD-inlet.