# Peer review of "Influence of ambient NO and NO2 on the quantification of total peroxy nitrates ( $\sum PNs$ ) and total alkyl nitrates ( $\sum ANs$ ) by thermal dissociation cavity ring-down spectroscopy (TD-CRDS)"

_EGUsphere, 2024_

## Author Response (AR1)

We sincerely thank the reviewers for their positive assessment of our manuscript and for their careful and thoughtful comments, to which we have responded point by point below. Comments are in black, our replies in blue and changes to the manuscript are in red.

**Reviewer 1 (Hans Osthoff):**

Wüst et al. provide an update on the quantification of total peroxy ($\Sigma$PN) and total alkyl nitrate ($\Sigma$AN) by thermal-dissociation cavity ring-down spectroscopy (TD-CRDS). New data are presented to show: (1) It is advantageous to oxidize NO to $NO_2$ by adding (excess) $O_3$ to the inlet after the thermal dissociation region. (2) Sampling synthetic PAN from a diffusion source will result in less inlet bias than sampling photochemically generated PAN (in the $\Sigma$AN channel). (3) Limonene derived nitrates suffer from different inlet biases than the commercially available isopropyl nitrate. (4) In high concentration, acetone generates an artifact in the $\Sigma$AN channel.

Regarding point (1): The authors do a good job convincing of the need to add $O_3$ after the TD region (and measure everything as $NO_x$) but this is not new. The first group to add $O_3$ after the TD region were, as far as I am aware, Wild et al. (Environm. Sci. Technol., 48(16), 9609-9615, doi:10.1021/es501896w, 2014). Our group has implemented the addition of $O_3$ since 2019 (see thesis by N. Gingerysty, doi: 10.11575/PRISM/38580 (2021)). Still, since there are groups quantifying $\Sigma$AN who are not adding $O_3$ (e.g., Lin et al. Talanta 270, 125524, 2024) it is worthwhile publishing this.

We agree and have added references to the Wild et al. paper: "An effective approach to remove the bias caused by NO oxidation to $NO_2$ has been described for TD-inlet systems designed to detect $NO_y$ via conversion to $NO_2$ (Friedrich et al., 2020; Wild et al., 2014)."

Regarding point (2): it is well known in the community that photochemical PAN sources co-emit organic by-products (especially formaldehyde) and are "pure" only relative to other nitrogen oxides. The data presented in this manuscript represent a near-worst-case scenario for a photochemical source as a very large acetone concentration and a Hg pen ray lamp were used. I agree with the author's conclusion that PAN from a diffusion source should be used to characterize $\Sigma$AN inlet chemistry instead. For the $\Sigma$PN channel, photochemical PAN sources are fine, though, as long as acetone concentrations are kept low.

Indeed, the use of high acetone (and thus high radical production rates and concentrations) make the situation worse. We have already cited the previous work of the Osthoff group which describes the problems in detail (at least for CIMS detection pf PAN) "The presence of organic acids and their possible interference in chemical-ionisation mass spectrometric studies of PAN has been documented previously (Mielke and Osthoff, 2012)"

(3) The data on limonene derived nitrates are novel and interesting. I was wondering if the difference to isopropyl nitrate is partially caused by a matrix effect (see specific comments).

We have addressed this in the "specific comments" section, but basically we are not surprised that the degradation of a C10 and a C3 organic nitrates leads to different amounts of $RO_2$ and $HO_2$ which can oxidize NO to $NO_2$.

I would have liked to see an experiment in which PAN (from the diffusion source) is combined with the chamber/limonene output to verify that 2+2 indeed equals 4 (and not 3.5 or 2.5) in both the ΣPN and ΣAN channels.

We have performed such experiments previously (Thieser et al., 2016). However, working with a mixture of PAN and ANs would NOT have shed light onto the ANs detection when using $NO_X$ detection as neither recombination nor conversion of NO to $NO_2$ play a role. In the PANs inlet, the ANs do NOT dissociate (if the tubing is made of PFA and not quartz) and thus there can be no impact on the recombination of radicals by adding what is effectively an inert gas. Conversely, PAN cannot cause problems in the ANs channel as the high temperatures prevent recombination of $NO_2$ with $CH_3C(O)O_2$ (and thus loss of $NO_2$).

(4) As an aside: We have observed a similar and more pronounced effect with acetone in our TD-CRDS $NO_y$ inlet (operated at ~850 K). Photochemical PAN sources are definitively no good for that application either!

Thank you for confirmation that photochemical PAN sources have their issues!

The paper is written well and very thoughtful. I recommend publication of this article once the authors have addressed my comments (above and below).

**Specific comments**

Line 14. Correct grammar ("show that the commonly used commercial C3-alkylnitrate (isopropyl nitrate, IPN) inlet for characterising")

Correction made.

Line 18. Please rephrase "We also show that using a photochemical source of PAN to characterise the TD-inlets can result in a much stronger apparent bias from NO to $NO_2$ conversion …". As commented on lines 150 – 151, the photochemical source used in this work could have been operated more optimally such that the statement in the abstract comes across as too strong.

Agreed. We now write: We also show that using a photochemical source of PAN to characterise the TD-inlets can result in a much stronger apparent bias from NO to $NO_2$ conversion than for a diffusion source of synthesised ("pure") PAN at similar mixing ratios, especially if high acetone concentrations (and thus radical concentrations) are involved.

Lines 90 - 91. "partial detection of ANs in the PNs channel" It is suggested that glass beads catalyze (i.e., lower the temperature at which) ANs convert to $NO_2$. Please comment on the potential effect of temperature gradients within the inlet (which can also lead to some overlap of the dissociation profiles and hence bias in TD-CRDS measurements of PNs and ANs).

Gradients in the temperature along the heated inlet will indeed "broaden" the dissociation profile. However, as long as all temperatures along the tube are lower than the measured one (measured at the heating wire on the outside of the tube) it is difficult to see how a temperature gradient can enable ANs detection in the PNs channel (the required temperatures are separated

by ~ 200 °C. It is however intuitive that surfaces catalyse the thermal dissociation of trace gases. Indeed, experiments in which (gas-phase) thermal dissociation rate coefficients are reported often have to passivate surfaces to avoid this.

Line 132. "effective absorption cross section". Consider stating its value (and how it was determined).

We have added text to describe this: "where $k_0$ and $k$ are the decay rate coefficients without and with absorbing/scattering medium, c is the speed of light and $\sigma_\lambda$ is the effective absorption cross section resulting from the overlap of the laser emission and the $NO_2$ absorption spectrum (Vandaele et al., 1998). The effective cross-section was obtained by regularly measuring the laser spectrum using a dedicated CCD camera (Ocean Optics HR4000) and was typically close to $6.4 \times 10^{-19}$ cm$^2$ molecule$^{-1}$."

Line 150. "4.6 % acetone". Flocke et al. (2005) "used a mixture of 10 ppmv acetone and 10 ppmv of CO … to keep the OH mixing ratio in the reaction vessel sufficiently low to almost completely suppress the formation of $HNO_3$, but … found that using 20 ppmv acetone instead was acceptable". If 4.6% acetone was used, it would not be surprising that this PAN photosource would contain a lot of impurities and result in bias.

We agree. The concentration of PAN in the source was adjusted to ensure that NO was converted to $NO_2$ and $NO_2$ to PAN. Too little acetone (and thus too low radical densities) limits the amount of NO that can be used and thus the PAN concentration that can be achieved. We chose high levels of NO and acetone to be able to vary the PAN concentration over a large range. As discussed in the paper, we recognize that such high radical concentrations can lead to "unwanted" secondary products and do not recommend this for either calibration of PAN instruments or (as in this case) tests of inlet chemistry. We now write: "In summary, use of a photochemical PAN source using high levels of acetone and high radical densities is not well suited for characterising the inlet chemistry for alkyl nitrate detection via thermal decomposition to $NO_2$ as it results in significant NO to $NO_2$ conversion owing to the presence of thermally labile trace gases such as peracetic acid and hydrogen peroxide. Used at high acetone concentrations they are also likely to cause matrix effects when deployed as a calibration source for mass-spectrometric detection of PAN. Indeed, the presence of organic acids and their possible interference in chemical-ionisation mass spectrometric studies of PAN has been documented previously (Mielke and Osthoff, 2012)."

Line 151. Phosphor-coated pen-ray lamps can get quite hot (which destroys PAN) and emit much radiation at 254 nm also that may drive unwanted photochemistry (see Furgeson et al. Atmos. Environm.45, 5025,doi: 10.1016/j.atmosenv.2011.03.072, 2011; Rider et al. Atmos. Meas. Tech. 8(7), 2737-2748, doi 10.5194/amt-8-2737-2015, 2015).

This is correct. We now cite the paper of Rider et al, who show how LEDs can be used to generate PANs. "PAN is formed at a yield of ~ 90-95 % (relative to NO consumed) by the sequence of reactions (R16-R19). The yield may be improved by using UV-LED sources that do not generate heat and thus cause PAN to thermally dissociate (Rider et al., 2015)."

Line 175. Correct spelling (tridecane). A safety note regarding the potential explosive nature of PAN should be added also.

Good idea ! We have added: "Note that safety precautions must be taken when working with samples of PAN, which has been known to explode under certain circumstances (e.g. when distilled from its tridecane solvent)." The spelling of tridecane has been corrected.

Line 192. Please change (CAN in 6 M HNO₃) to (CAN) in 6 M HNO₃ to improve clarity.

Correction made

Line 200. "ANs mixing ratios between 0.47 and 4.4 ppbv". How was this determined?

At this point we wish merely to state approximately how much AN was generated using this methods which is the CRD-measured value at zero added NO. We now write: Isoprene and limonene were introduced into the chamber via flow-controllers attached to steel canisters containing isoprene (98%, Acros Organics, 45 ppmv in He) or limonene (96%, Sigma-Aldrich, 44 ppmv in He) in He (5.0, Westfalen), resulting in ANs mixing ratios up to $\approx 4$ ppbv (see later).

Line 201. "The nitrates generated in this manner from isoprene are a mixture of C5-nitrooxyhydroperoxides, C5-nitrooxycarbonyl, C5hydroxynitrate and also C10-nitrooxyperoxide (ROOR), with the relative concentrations depending on the fate of the initially formed nitrooxyperoxy radicals (IUPAC 2024)." I didn't find IUPAC in the reference list (only in the SI) - there may be a more suitable reference for this statement anyways.

We now list references to an extensive experimental dataset (Schwantes et al., 2015) as well as two reviews (inc. IUPAC) that describe the products formed. IUPAC is now correctly cited in the bibliography. We write: "The nitrates generated in this manner from isoprene are a mixture of C5-nitrooxyhydroperoxides, C5-nitrooxycarbonyl, C5-hydroxynitrate and also C10-nitrooxyperoxide (ROOR), with the relative concentrations depending on the fate of the initially formed nitrooxyperoxy radicals (Schwantes et al., 2015; Ng et al., 2017; Iupac, 2024).

Lines 239-310 "At 648 K the chemistry is different". What follows is interesting descriptions of possible reaction pathways. What is missing, though, are numerical simulations to constrain the impact of each of these side reactions. For example, I would be curious how much of the singlet α-lactone forms via (R29) and what the impact of this pathway could be (i.e., by how much does the product distribution shift when this reaction is "turned off" in the mechanism?). In other words, this section could be more quantitative (and thus appear less speculative).

This chemistry in the hot inlet is thoroughly documented by Thieser et al. (2016) and Sobanski et al. (2016) (and references therein) and is based on published experimental datasets. However, as described by Thieser and Sobanski there are many uncertainties we did not strive to generate a reaction mechanism (inc. both gas phase and heterogeneous processes) that perfectly mimics the experimental data as too many parameters (including the formation and further reactions of the α-lactone) are poorly characterised at the relevant temperature. We prefer not to repeat this here and lengthen the manuscript unnecessarily with repeated discussions. Indeed,the whole point of this manuscript is to show that such modelling can be avoided in NO$_X$ instead of NO$_2$ is measured.

Line 310. "In separate experiments, we observed that the thermal dissociation of acetone also accounts for a small fraction of the overestimation of NO₂ formed (400 pptv NO₂ at 43 ppmv acetone and 16 ppbv NO), which is confirmed by its thermogram (Fig. S2)." Carbon, hydrogen and oxygen atoms do not convert to nitrogen (and form nitrogen dioxide), so an explanation is

needed what causes this effect. My hunch is that heating acetone to these temperatures generates α,β-dicarbonyls (methyl glyoxal or 2,3-butadione) which would absorb at 409 nm.

The NO$_2$ is formed by the thermal dissociation of acetone in air to form peroxy radicals that convert NO to NO$_2$. The presence of NO was perhaps not made clear enough. We now write: "In separate experiments, we observed that (in the presence of NO) the thermal dissociation of acetone also accounts for a small fraction of the excess of NO$_2$ formed (400 pptv NO$_2$ at 43 ppmv acetone and 16 ppbv NO), which is confirmed by its thermogram (Fig. S2). This is presumably a result of surface calalysed dissociation of acetone to form (in the presence of air) peroxy radicals that convert NO to NO$_2$."

Line 322. "In summary, the photochemical PAN source is not well suited for characterising the inlet chemistry for alkyl nitrate detection via thermal decomposition to NO$_2$ as it results in significant NO to NO$_2$ conversion owing to the presence of thermally labile trace gases such as peracetic acid and hydrogen peroxide". What if a lower acetone concentration had been used?

Good point! We have modified the text to write: "In summary, use of a photochemical PAN source using high levels of acetone and high radical densities is not well suited for characterising the inlet chemistry for alkyl nitrate detection via thermal decomposition to NO$_2$ as it results in significant NO to NO$_2$ conversion owing to the presence of thermally labile trace gases such as peracetic acid and hydrogen peroxide. Used at high acetone concentrations the photochemical source is likely to cause matrix effects when deployed as a calibration source for mass-spectrometric detection of PAN. Indeed, the presence of organic acids and their possible interference in chemical-ionisation mass spectrometric studies of PAN has been documented previously (Mielke and Osthoff, 2012). This can be alleviated by using much lower acetone concentrations (and thus radical production rates) which however limits the amount of PAN that can be generated."

Line 330. "The IR spectrum (Fig. S4) of a gas sample eluted from the PAN diffusion source is in good agreement with the literature and does not reveal the presence of high levels of impurities." Please add more information (to the experimental section or the SI) how this spectrum was obtained.

We have added details to the Figure caption: "Intensity-normalised FTIR spectrum (Bruker Vector 22, 0.5 cm$^{-1}$ resolution, 128 scans) of PAN in air at room temperature recorded in a 45 cm optical absorption cell at a total pressure of ~ 2 Torr . Good agreement with the spectral features reported by Allen et al. (2005) (black symbols) is observed. PAN was transported into the absorption cell by flowing N$_2$ over the sample held at 0 °C in tridecane."

Line 375. "0.47 or 4.4 ppbv isoprene nitrate" How were these mixing ratios determined?

These mixing ratios were not determined using separate instrumentation but are derived from the mixing ratio observed by the CRD at zero added NO. The point being made here is that there is an increase in the NO$_2$ signal when detecting NO$_2$ but not when detecting NO$_X$ and that this is applicable at both high and low mixing ratios of the ANs generated. In order not to make the impression that the mixing ratios were known BEFORE entering the 5ch-CRD, we have given approximate values instead of those previously listed (e.g. we now state ~ 0.5 or ~4 ppbv instead of 0.47 and 4.4 ppbv.)

Line 417-420. "when using biogenic nitrates derived from terpenoids." The authors sampled biogenic nitrates in a complex mixture containing all kinds of compounds. The conclusions drawn may thus be more due to a matrix effect which would be absent when sampling IPN. In other words, would these results stand if a "pure" sample of limonene nitrate (isolated or synthesized) had been analyzed?

This is an interesting point, which (as we cannot make "pure" limonene nitrates) we cannot explore experimentally. However, the fact that longer chain biogenic nitrates can result in larger effects is intuitive and indeed expected if the carbon backbone sequentially breaks down at high temperatures to form peroxy radicals. The experiment in which $NO_2$ was detected while working with limonene nitrate (Figure 7b) was conducted with only ~1ppbv of the nitrate present, which is not significantly higher than that which might be found in the atmosphere. As we detect only $NO_2$ we do not expect matrix effects to influence the detection sensitivity (as see e.g. in the CIMS experiments of Mielke et al).

Figure S1. Please state in the caption for what temperature the simulations were run.

Done.

**Reviewer 2:**

**General comments:**

This paper describes a thorough set of experiments to quantify the effect of additional (i.e., other than that produced by thermal dissociation) NO and NO2 on the performance of a TD-CRDS instrument for the measurements of peroxy- and alkyl nitrates. It also provides a helpful overview of other studies that have characterized this secondary inlet chemistry, which serves as a nice review of the technique. This will be of great interest to the community using such thermal dissociation techniques for organic nitrate measurements. The paper is generally very well-written and clear. I recommend publication after some minor revisions to clarify especially the figures.

**Specific comments:**

Several times throughout the paper you use the term "eluted" for transfer of a liquid into the gas phase. I think "evaporated" would be more accurate; "eluted" typically refers to a separation of liquids.

We have replaced the term "eluted" with "evaporated" throughout the manuscript.

Around lines 287-288: This is a very large range of thermal dissociation lifetimes! Maybe comment here on how well known these are, and how this means the organic composition in ambient samples will heavily influence the secondary chemistry. This can help anticipate the uncertainties in the simulations you show later.

The thermal dissociation lifetimes are only used to illustrate that the peroxides considered all decompose to some extent in the heated inlet at 648 K. For the organic peroxides this may be a gas-phase process, for $H_2O_2$ it is likely to be accelerated by the surface. We have adjusted the text as follows: "These trace gases are stable at room temperature but both $CH_3OOH$ and $CH_3C(O)OOH$ have lifetimes with respect to thermal dissociation of the order of seconds at ~ 650 K. (Sahetchian et al., 1992; Kirk, 1965). In contrast, the thermal lifetime of $H_2O_2$ at 650 K is several minutes (Baulch et al., 2005) which strongly suggests that its decomposition in our heated inlet is catalysed by the quartz tubing."

The associated rate coefficients were NOT used in any simulations. The simulations of the photochemical PAN source, which the reviewer may be referring to, were performed at room temperature and serve only to show that such peroxides are present.

Around line 302: You start using PAA later in this paragraph, so nice to introduce that acronym in the first line.

The acronym 'PAA' has now been introduced in the first line of the paragraph.

In the paragraph on lines 313-321, you describe the model simulations shown in Fig. S3. Since you do some experiments adding specific peroxides, can you comment on how well these simulated patterns match with your observations? Also, can you comment on how you arrived at the (very high) concentrations used in these simulations?

We did not attempt to simulate the thermal dissociatuion of the peroxides. The simulations shown in Fig S3 show the (undiluted) output of the photochemical reactor (and are thus very high). The dilution factor is $\approx 1/60$, which reduces the PAN level in the CRD to about 1.5 ppbv, as observed. The peroxides will be diluted to the same extent. Some text has been added to the caption of Fig S3 to clarify this. "The mixing ratios at 120 s represent those at the exit of the reactor prior to dilution (factor ~ 60) and before entering the heated inlets and cavities. The reaction scheme (all rate coefficients calculated for room temperature) used is listed in Table S1." The simulations of the reactor without NO (i.e. without PAN formation) have been removed in order to avoid overcomplication of this section of the manuscript.

On line 337, you refer to "impurities" when I think you mean "additional organic species that can oxidize NO to NO2"

Agreed. We now write: "Compared to the photochemical PAN source, the bias ($NO_2$ formed per ppbv NO added) is smaller, which is related to the absence of thermally unstable products such as $CH_3C(O)OOH$."

On line 359 "cause by the oxidation of NO2" – here, don't you mean the recombination of NO2 with organics?

Yes. We have changed the sentence to: "Thus, numerical simulations are still required to correct for the bias caused by the recombination of the acetyl peroxy radical with $NO_2$."

On line 373 you refer to "more relevant (unsaturated) alkyl nitrates". This makes me wonder if the fact that first-generation terpenes alkyl nitrates are unsaturated is the key here? Below around line 390 you discuss the degradation of longer chain alky nitrates producing more organic peroxy radicals. Do you think the double bond plays a role here, or is it just carbon number?

We do not know which molecular characteristics are the most relevant for the number of $RO_2$ sequentially formed following the initial thermal degradation of the nitrate. This will depend both on the chain length (which ultimately determines the maximum number of $RO_2$ and thus $HO_2$ that can be formed) and bond strengths. Whether the degree of chain degradation within the heated inlets is expedited by the presence of unsaturated bonds is not known and is likely to be highly variable for different organic precursors. This underlines the importance of measuring $NO_X$ rather than $NO_2$.

On line 405 you say this means unreliable data unless NO levels are very low. Can you add a comment based on your ambient measurements how often / where this is case? I wonder if this is mainly a problem for urban / near-source datasets?

Urban environments could certainly be a problem. We now comment on this: "This would limit the usefulness of such devices to locations with NO < $\approx$ 2 ppbv) (Sobanski et al., 2016) as found in remote/rural environments or nighttime when NO levels are often low unless nearby emission sources are present (as e.g. in urbanized regions)."

Figure 2: is a bit confusing because the lower panel is without NO in the PAN source, but you see the NO axis based on the added NO in the cavities. Maybe modify the last sentence to clarify that you in this case add only the organics and not the NO from the PAN source? The y-axis error bars are only visible on the panel b 648 K trace. Are they present on all of them and

just so small to be invisible? If so, state this. But I don't know why this would be so different e.g. form panel a to b 648 K traces.

In order to remove ambiguity (and explain the error bars), the Figure caption has been changed to: **"Figure 1:** $NO_2$ measured in the cavities with inlet temperatures of 448 K and 648 K in the presence of different amounts of NO with (a) and without (b) PAN present. The error bars in the x-direction include uncertainty in the mass flow controllers (1 % of the maximum flow of the MFC) and the uncertainty in the mixing ratio of the NO bottle (5 %). The error bars in the y-direction (smaller than the symbols in panel b) include the standard deviations (1 σ) of the $NO_2$ mixing ratios both at 648 K and at room temperature."

Finally, you refer to the errors including 1sigma NO2 mixing ratio std deviations of both 648 K and room temperature NO2 mixing ratios. Does this mean you subtract the room temperature measurements in each case? That didn't seem to be the case to me, since you're not reporting sumANs but rather NO2 in the 648 K channel (sounds like it would be an unsubtracted quantity). And, do you do this subtraction only for the 648 k trace and not the 448K trace, since you don't mention the latter? This caption raises a lot of questions that I think some revision could avoid. In the later figures you also refer to the uncertainty in room temperature NO2 measurements. Maybe add to each caption what you subtract to get each trace, to clarify.

We write (L221-L222) "For both datasets, the $NO_2$ measured in the room temperature cavity (present as an impurity in the PAN sample) has already been subtracted." This (or similar text) is now added to each Figure caption, where appropriate.

For parallel structure, could the x axis of Figure 6 be listed as "NO2 added (ppbv)"?

Yes. Done.

**Technical corrections:**

Line 166: "In section 4.1, we discuss the concentrations"

Correction made.

Line 175: "tridecan**e**"

Correction made.

Line 218: "so that the cavities detected **only** NO2 (and not NOx)."

Correction made.

Line 311: "fraction of the excess of NO2 formed"

Correction made.

Line 317: "S3a), 88 ppbv PANs"

"88 ppbv PAN" is correct (as written).

Line 320: "Fig. S3b), higher mixing"

Correction made.

Line 365: "we have shown **that** IPN"

Correction made.

Line 416: "we have **quantified** the potential bias"

Correction made.

Line 420: "as has been frequently **assumed**."

Correction made.

Line 428 (a bit long sentence, maybe reword? Or at least): "to convert NO to NO2**,** corrections for"

Correction made.

**References:**

Allen, G., Remedios, J. J., Newnham, D. A., Smith, K. M., and Monks, P. S.: Improved mid-infrared cross-sections for peroxyacetyl nitrate (PAN) vapour, Atmos. Chem. Phys., 5, 47-56, 10.5194/acp-5-47-2005, 2005.

Baulch, D. L., Bowman, C. T., Cobos, C. J., Cox, R. A., Just, T., Kerr, J. A., Pilling, M. J., Stocker, D., Troe, J., Tsang, W., Walker, R. W., and Warnatz, J.: Evaluated kinetic data for combustion modeling: Supplement II, Journal of Physical and Chemical Reference Data, 34, 757-1397, 2005.

Friedrich, N., Tadic, I., Schuladen, J., Brooks, J., Darbyshire, E., Drewnick, F., Fischer, H., Lelieveld, J., and Crowley, J. N.: Measurement of NOx and NOy with a thermal dissociation cavity ring-down spectrometer (TD-CRDS): instrument characterisation and first deployment, Atmos. Meas. Tech., 13, 5739-5761, 10.5194/amt-13-5739-2020, 2020.

IUPAC Task Group on Atmospheric Chemical Kinetic Data Evaluation, (Ammann, M., Cox, R.A., Crowley, J.N., Herrmann, H., Jenkin, M.E., McNeill, V.F., Mellouki, A., Rossi, M. J., Troe, J. and Wallington, T. J.). Last access April. 2024: https://iupac.aeris-data.fr/, last

Kirk, A. D.: The Thermal Decomposition of Methyl Hydroperoxide, Can. J. Chem., 43, 2236-2242, 1965.

Mielke, L. H. and Osthoff, H. D.: On quantitative measurements of peroxycarboxylic nitric anhydride mixing ratios by thermal dissociation chemical ionization mass spectrometry, International Journal of Mass Spectrometry, 310, 1-9, https://doi.org/10.1016/j.ijms.2011.10.005, 2012.

Ng, N. L., Brown, S. S., Archibald, A. T., Atlas, E., Cohen, R. C., Crowley, J. N., Day, D. A., Donahue, N. M., Fry, J. L., Fuchs, H., Griffin, R. J., Guzman, M. I., Herrmann, H., Hodzic, A.,

Iinuma, Y., Jimenez, J. L., Kiendler-Scharr, A., Lee, B. H., Luecken, D. J., Mao, J., McLaren, R., Mutzel, A., Osthoff, H. D., Ouyang, B., Picquet-Varrault, B., Platt, U., Pye, H. O. T., Rudich, Y., Schwantes, R. H., Shiraiwa, M., Stutz, J., Thornton, J. A., Tilgner, A., Williams, B. J., and Zaveri, R. A.: Nitrate radicals and biogenic volatile organic compounds: oxidation, mechanisms, and organic aerosol, Atmospheric Chemistry and Physics, 17, 2103-2162, 10.5194/acp-17-2103-2017, 2017.

Rider, N. D., Taha, Y. M., Odame-Ankrah, C. A., Huo, J. A., Tokarek, T. W., Cairns, E., Moussa, S. G., Liggio, J., and Osthoff, H. D.: Efficient photochemical generation of peroxycarboxylic nitric anhydrides with ultraviolet light-emitting diodes, Atmospheric measurement techniques, 8, 2737-2748, 10.5194/amt-8-2737-2015, 2015.

Sahetchian, K. A., Rigny, R., Tardieu de Maleissye, J., Batt, L., Anwar Khan, M., and Mathews, S.: The pyrolysis of organic hydroperoxides (ROOH), Symposium (International) on Combustion, 24, 637-643, https://doi.org/10.1016/S0082-0784(06)80078-0, 1992.

Schwantes, R. H., Teng, A. P., Nguyen, T. B., Coggon, M. M., Crounse, J. D., St Clair, J. M., Zhang, X., Schilling, K. A., Seinfeld, J. H., and Wennberg, P. O.: Isoprene $NO_3$ Oxidation Products from the $RO_2 + HO_2$ Pathway, Journal of Physical Chemistry A, 119, 10158-10171, 10.1021/acs.jpca.5b06355, 2015.

Sobanski, N., Schuladen, J., Schuster, G., Lelieveld, J., and Crowley, J. N.: A five-channel cavity ring-down spectrometer for the detection of $NO_2$, $NO_3$, $N_2O_5$, total peroxy nitrates and total alkyl nitrates, Atmospheric Measurement Techniques, 9, 5103-5118, 10.5194/amt-9-5103-2016, 2016.

Thieser, J., Schuster, G., Phillips, G. J., Reiffs, A., Parchatka, U., Pöhler, D., Lelieveld, J., and Crowley, J. N.: A two-channel, thermal dissociation cavity-ringdown spectrometer for the detection of ambient $NO_2$, $RO_2NO_2$ and $RONO_2$, Atmos. Meas. Tech., 9, 553-576, 10.5194/amt-9-553-2016, 2016.

Wild, R. J., Edwards, P. M., Dube, W. P., Baumann, K., Edgerton, E. S., Quinn, P. K., Roberts, J. M., Rollins, A. W., Veres, P. R., Warneke, C., Williams, E. J., Yuan, B., and Brown, S. S.: A measurement of total reactive nitrogen, NOy, together with $NO_2$, NO, and $O_3$ via cavity ring-down spectroscopy, Environmental Science & Technology, 48, 9609-9615, doi:10.1021/es501896w, 2014.